# FLARE: Fast Low-rank Attention Routing Engine

## Abstract

The quadratic complexity of self-attention limits its applicability and scalability on long sequences. We introduce *Fast Low-rank Attention Routing Engine (FLARE)*, a simplified latent-attention mechanism wherein each attention head encodes $N$ input tokens onto $M \ll N$ latent tokens and immediately deocdes back. This produces an implicit rank $\leq M$ attention operator using only two cross-attention steps providing a simple and mathematically transparent low-rank formulation of self-attention that can be implemented in $\mathcal{O}(MN)$ time. Crucially, FLARE eliminates latent-space self-attention and, by expressing both encode and decode steps as fused-attention operations, avoids ever materializing the $M \times N$ projection matrices. Moreover, by assigning each head independent latent token slices, FLARE realizes multiple parallel low-rank pathways that collectively approximate a richer global attention pattern without sacrificing efficiency. Empirically, FLARE trains end-to-end on *one-million-point* unstructured meshes on a single GPU, achieves state-of-the-art accuracy on PDE surrogate benchmarks, and outperforms general-purpose efficient-attention methods on the Long Range Arena suite. We additionally release a large-scale additive manufacturing dataset to spur further research.

## 1 Introduction

High-fidelity simulations of physical systems are often too costly for multi-query applications, such as design optimization or uncertainty quantification. Machine learning offers a promising alternative via surrogate models that learn system dynamics from data, enabling fast approximations that accelerate experimentation and decision making.

Among machine learning (ML) architectures, transformers (Vaswani et al., 2017) have shown exceptional scalability and generalization capabilities in domains such as natural language processing (Devlin et al., 2019), computer vision (Dosovitskiy et al., 2020) and beyond. This has motivated adapting transformers to spatially distributed data such as point clouds and meshes in physical simulations, where each mesh point is treated as a token with features encoding geometry and physical fields. However, applying transformers directly to large-scale unstructured meshes introduces severe computational bottlenecks: for a sequence of $N$ tokens, standard self-attention scales as $\mathcal{O}(N^2)$ in time and memory. While this global communication is key to the expressive power of transformers, and helps them outperform inherently local graph neural networks (Yun et al., 2019), its quadratic footprint hinders scalability.

To alleviate the quadratic cost, several architectures reduce self-attention to $\mathcal{O}(NM)$ complexity by introducing a latent dimension $M \ll N$. As these models restrict attention to an $M$-token latent bottleneck, they represent strict subsets of full self-attention. Consequently, their expressivity is capped by the latent dimensionality, and they cannot theoretically outperform vanilla self-attention beyond minor regularization-related gains.

Perceiver and PerceiverIO (Jaegle et al., 2021b;a) introduce an attention-based projection between the input sequence and a learnable latent sequence of length $M$. Perceiver uses $M \times N$ cross-attention to encode $N$ input tokens into $M$ latents, followed by latent self-attention. Optional additional cross-attention layers can pull information from the input back into the latent space, helping recover details not fully captured during encoding. This reduces the cost of full self-attention on inputs but introduces explicit projection modules and repeated latent-space operations. These repeated

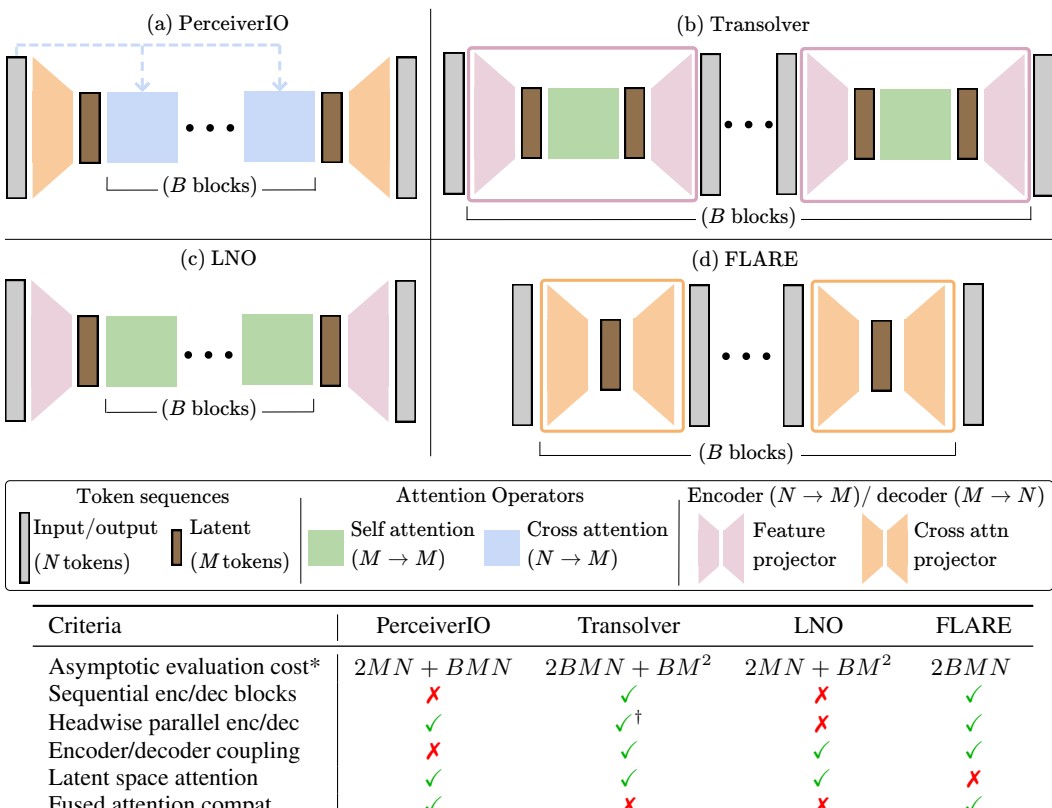

| Criteria | PerceiverIO | Transolver | LNO | FLARE |
|---|---|---|---|---|
| Asymptotic evaluation cost* | $2MN + BMN$ | $2BMN + BM^2$ | $2MN + BM^2$ | $2BMN$ |
| Sequential enc/dec blocks | ✗ | ✓ | ✗ | ✓ |
| Headwise parallel enc/dec | ✓ | ✓[†] | ✗ | ✓ |
| Encoder/decoder coupling | ✗ | ✓ | ✓ | ✓ |
| Latent space attention | ✓ | ✓ | ✓ | ✗ |
| Fused attention compat. | ✓ | ✗ | ✗ | ✓ |

*Evaluating cost of processing $N \gg M$ tokens ignoring feedforward layers (FFN are $\mathcal{O}(N)$).

[†]Physics Attention in Transolver uses the same projection weights for all heads, whereas PerceiverIO and FLARE use cross-attention projection where each head learns a distinct slice of the latent tokens.

Figure 1: Comparison of (a) PerceiverIO (Jaegle et al., 2021a), (b) Transolver (Wu et al., 2024), (c) LNO (Wang & Wang, 2024b), and (d) FLARE (ours). Each model contains $B$ blocks and $M$ latents. The criteria in the table are described in detail in Appendix A.

latent transformations make it challenging to characterize the resulting token-to-token communication pattern or to reason about the effective low-rank structure of the model.

In PDE surrogate modeling, Transolver (Wu et al., 2024) and Latent Neural Operator (LNO) (Wang & Wang, 2024a) adopt related projection-based schemes to map variable-length point clouds to fixed-length latent representations. Both obtain projection weights by expanding the feature dimension and then applying self-attention on the resulting latent sequence. Transolver further performs projection and unprojection in every transformer block, forming deep models where downstream layers can exploit global context aggregated by earlier blocks. However, the feature projection layers in Transolver and LNO cannot be implemented purely via standard scaled dot-product attention (SDPA) kernels (Dao et al., 2022): they explicitly construct or operate on $M \times N$ projection weights, which becomes a memory bottleneck on long sequences.

Figure 1 summarizes these designs and their asymptotic costs. PerceiverIO and LNO perform a single encode–decode step with latent self-attention; Transolver performs repeated projection/unprojection plus latent self-attention in each block. Transolver and LNO obtain projection weights by feature expansion, whereas Perciever encodes via multihead cross attention projection.

We propose **Fast Low-rank Attention Routing Engine (FLARE)**, a simple yet expressive mechanism designed to overcome the scalability limitations of transformer-based models. FLARE simplifies the latent attention bottleneck paradigm by noting that encoding an input sequence onto $M$ latent tokens and immediately decoding back yields an implicit rank $\leq M$ formulation of self-attention. This stands in contrast to prior latent-token architectures, which rely on deep computations within the latent space and therefore obscure the mathematical structure of the induced attention map. In

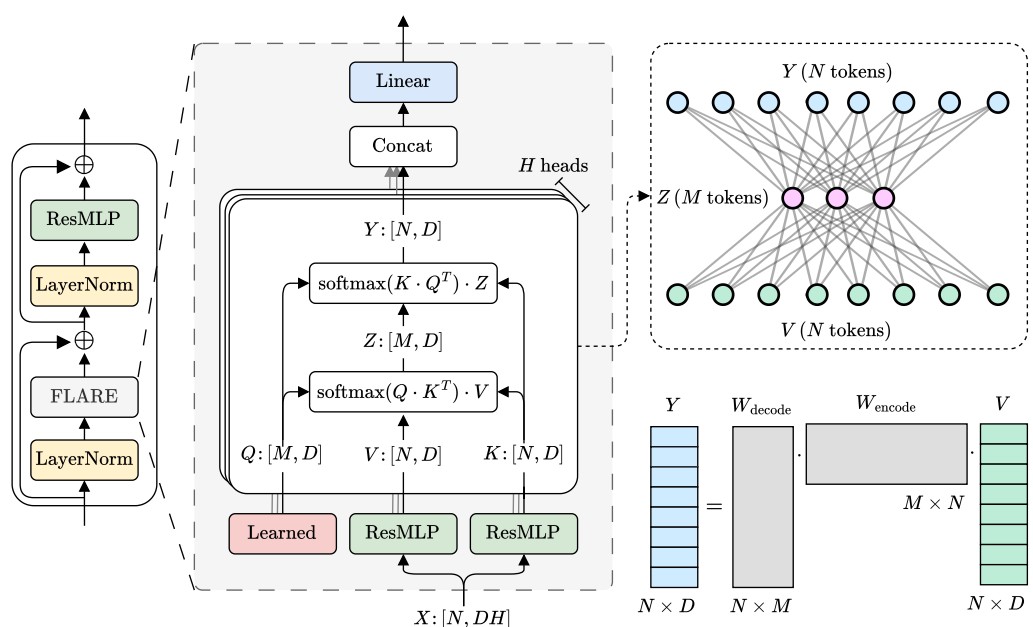

Figure 2: Schematic of a flare block. In flare, each head projects the input sequence with $n$ tokens to a fixed-length sequence of $m$ tokens via the cross-attention matrix $W_{\text{encode}} = \text{softmax}(Q \cdot K^T)$, and then projects back to the original length via the cross-attention matrix $W_{\text{decode}} = \text{softmax}(K \cdot Q^T)$. The overall operation is equivalent to token mixing on the input sequence with the rank-deficient matrix $(W_{\text{decode}} \cdot W_{\text{encode}})$.

FLARE, both the encode and decode steps are implemented using standard SDPA kernels, enabling the full $\mathcal{O}(NM)$ operation to run efficiently without materializing the $M \times N$ projection matrices.

A defining structural property of FLARE is the elimination of latent-space self-attention: each block consists solely of two cross-attention operations. This yields a single, homogeneous attention block that can be stacked to arbitrary depth while maintaining linear complexity. Another central intuition behind this design is that each attention head acts as an independent low-rank projection–reconstruction pathway. When many such pathways operate in parallel, the model learns a mixture of low-rank components that collectively approximate a richer global communication pattern than any individual projection could capture.

To enable this behavior, FLARE assigns each head its own slice of latent tokens, yielding distinct projection matrices per head and allowing heads to specialize in complementary routing patterns. Unlike Transolver, which shares projection weights across heads, or LNO, which uses a single global projection, FLARE's head-wise independence fosters greater representational diversity. To study this effect, we introduce a linear-time eigenanalysis method for the implicit low-rank attention matrices. Our spectral analysis reveals marked diversity in eigenvalue decay across heads, providing empirical evidence that independent latent slices support richer mixtures of rank $M$ factors than shared projections. Figure 2 illustrates a single FLARE block.

We summarize our main contributions below.

- **A unified low-rank formulation of self-attention.** We express self-attention as two cross-attention operations through a learnable latent sequence, yielding a rank $\leq M$ attention operator. This formulation eliminates latent-space self-attention, resulting in a single, homogeneous block with $\mathcal{O}(NM)$ complexity whose structure is mathematically transparent. Because this operator is composed purely of attention projections, it can be implemented using standard SDPA kernels as a consequence of the formulation.

- **Strong empirical performance across PDE surrogates and general long-range tasks.** Across multiple PDE benchmarks, FLARE achieves superior predictive accuracy compared to leading neural surrogate models, despite using fewer parameters. FLARE also surpasses general-purpose efficient-attention models on the Long Range Arena (LRA) (Tay et al.)

benchmark suite, demonstrating that our low-rank construction is broadly effective beyond PDE surrogate learning.

- **Scalability to million-point meshes.** Because all bottleneck operations are realized with fused attention primitives, FLARE attains high GPU utilization and scales to unstructured meshes with **one million points** on a single H100 80GB GPU, without distributed training or memory offloading, which is to our knowledge, the largest scale demonstrated for transformer-based PDE surrogates. This capability addresses a practical bottleneck in industrial-scale surrogate modeling and demonstrates the effectiveness of the proposed design at extreme input sizes.

- **Extensive architectural validation.** All major components of FLARE are supported by targeted ablations, which together provide causal evidence for the design choices underlying our low-rank formulation.

- **Benchmark dataset for additive manufacturing.** We release a large-scale, high-resolution dataset of thermomechanical additive manufacturing simulations for residual displacement prediction, providing a challenging benchmark for scalable PDE surrogate modeling and a reproducible testbed for future work.

## 2 RELATED WORK

**Neural PDE surrogates.** Learning neural surrogates for PDEs is an exciting and rapidly growing direction in scientific machine learning. Neural operators (Li et al., 2020; Lu et al., 2021; Kovachki et al., 2023) have been proposed for learning mappings between infinite-dimensional input and output function spaces, enabling mesh-independent generalization. Extensions to neural operators that incorporate geometric priors and graph-based representations further improve performance, especially for problems defined on complex unstructured meshes (Li et al., 2023b;c).

Graph networks have also been widely explored as PDE surrogates (Pfaff et al., 2020; Elrefaie et al., 2024; Ferguson et al., 2025), leveraging their inherent ability to model local neighborhood interactions on meshes. Transformer-based architectures have emerged more recently as powerful PDE surrogates (Alkin et al., 2024; Cao, 2021; Li et al., 2022; 2023a; Hao et al., 2023), allowing global context aggregation and dynamic prediction of complex phenomena (Qian et al., 2025). Recent works (Alkin et al., 2024; Wu et al., 2024; Wang & Wang, 2024a) leverage latent space attentions for PDE modeling, achieving high accuracy with reduced computational cost. Building upon these advances, our FLARE method employs a linear complexity attention mechanism with a learnable latent bottleneck to further improve accuracy and efficiency on PDE surrogate modeling tasks.

**Efficient attention mechanisms.** Several methods have been proposed to address the quadratic complexity of the attention mechanism. In (Wang et al., 2020), attention matrices are shown to be low-rank. Their proposed model, Linformer, learns $M \times N$ matrices that projects $N$ key/value tokens into $M$ latent tokens: $\tilde{K} = E_K \cdot K$, $\tilde{V} = E_V \cdot V$. While effective in some settings, this strategy has notable limitations. First, it assumes a fixed and ordered token structure, which may not be suitable for unstructured grids. Second, Linformer requires storing $\mathcal{O}(NM)$ parameters per block, which can become prohibitive for long sequences. For example, with $N = 1$m and $M = 512$, each Linformer block would require 512m parameters, making this approach impractical.

Other low-rank approaches include Reformer (Kitaev et al., 2020), which uses locality-sensitive hashing for efficient approximate attention, Nyströmformer (Xiong et al., 2021), which approximates self-attention using the Nyström method, and Scatterbrain (Chen et al., 2021), which combines sparse and low-rank methods for further gains. Another efficient attention method, called multi-head latent attention (Liu et al., 2024), jointly compresses key and value tokens to reduce the memory footprint of language models during inference. However, this does not help with the quadratic bottleneck in self-attention. Low-Rank Adaptation (LoRA) (Hu et al., 2022) also leverages low-rank projections, though primarily for efficient fine-tuning rather than architectural design or modeling capacity, as we explore in this work.

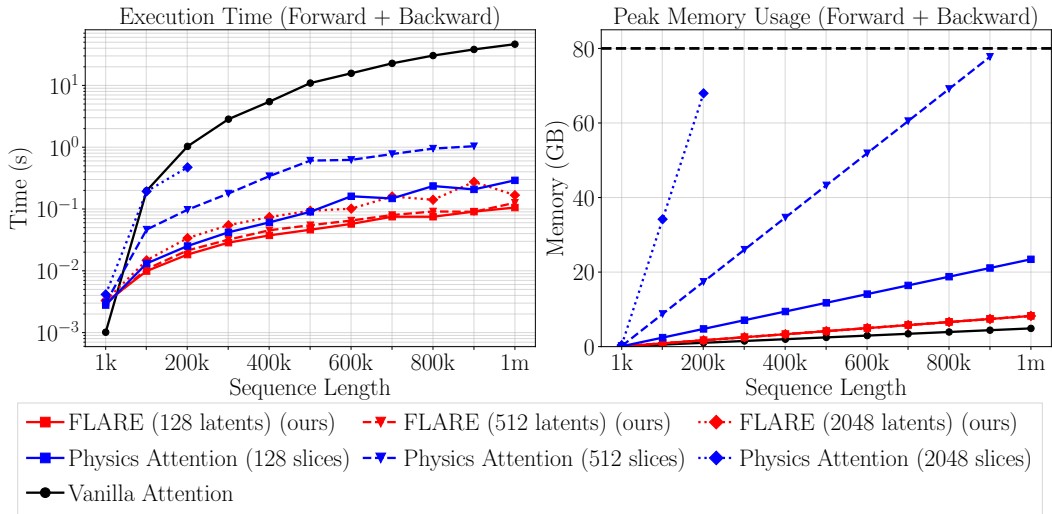

Figure 3: Time and memory requirements of different attention schemes. On an input sequence of one million tokens, FLARE (red) is over $200\times$ faster than vanilla attention, while consuming marginally more memory. All models are implemented with flash attention (Dao et al., 2022), and the memory upper bound on a single H100 80GB GPU is depicted with a dashed line. Note that the curves for FLARE are somewhat overlapping. A detailed analysis is presneted in Appendix G.

## 3 METHOD

### 3.1 PRELIMINARY: MULTI-HEAD SELF-ATTENTION

Let $X \in \mathbb{R}^{N \times C}$ denote the input sequence of $N$ tokens with $C$ features each. The query, key, and value matrices $Q, K, V \in \mathbb{R}^{N \times C}$ are obtained by applying learned linear projections to $X$, we have

$$Q = X \cdot W^q, \quad K = X \cdot W^k, \quad V = X \cdot W^v, \tag{1}$$

where $W^q, W^k, W^v \in \mathbb{R}^{C \times C}$. The $Q$, $K$, $V$ matrices are then split along the feature dimension and passed to $H$ heads, each with dimension $D = C/H$, enabling parallel computation of attention:

$$[Q_1, \ldots, Q_H] = Q, \quad [K_1, \ldots, K_H] = K, \quad [V_1, \ldots, V_H] = V. \tag{2}$$

The scaled dot-product attention (SDPA) operation, introduced by (Vaswani et al., 2017), computes the output as

$$Y_h = \text{SDPA}(Q_h, K_h, V_h, s) = \text{softmax}\left(\frac{Q_h \cdot K_h^T}{s}\right) \cdot V_h, \tag{3}$$

where $Q_h, K_h, V_h \in \mathbb{R}^{N \times D}$ are query, key, and value matrices belonging to head $h$, and $s$ is typically $\sqrt{D}$. Note that softmax is taken along the row-dimension. The outputs $Y_h$ from all heads are then concatenated along the feature dimension to form the final output

$$Y = [Y_1, \ldots, Y_H]. \tag{4}$$

This concatenation, followed by a linear layer, enables the model to integrate information across attention heads efficiently.

The greatest cost in multi-head self-attention is the call to SDPA which is $\mathcal{O}(N^2)$ in time and memory complexity. This is because $Q_h \cdot K_h^T \in \mathbb{R}^{N \times N}$ requires $\mathcal{O}(N^2)$ storage and softmax, matrix-vector product with $V_h$ takes $\mathcal{O}(N^2)$ operations. Fortunately, GPU optimized multi-head implementations of SDPA are available in PyTorch (Paszke, 2019).

### 3.2 FLARE: FAST LOW-RANK ATTENTION ROUTING ENGINE

FLARE is a linear-complexity token mixing layer that learns low-rank global communication structures via attention projections. The FLARE mechanism introduces a set of $M \ll N$ learnable latent tokens that serve as a bottleneck for information exchange. The process consists of two stages:

```
1  import torch.nn.functional as F
2  def flare_multihead_mixer(Q, K, V):
3      # Args - Q: [H, M, D], K, V: [B, H, N, D]
4      # Ret - Y: [B, H, N, D]
5      Z = F.scaled_dot_product_attention(Q, K, V, scale=1.0)
6      Y = F.scaled_dot_product_attention(K, Q, Z, scale=1.0)
7      return Y
```

Figure 4: PyTorch code for multi-head token mixing operation in FLARE. See Figure 7 for an implementation without the fused attention kernel.

1. **Encoding.** The input sequence is projected onto the latent tokens via cross-attention, compressing global information.

2. **Decoding.** The latent tokens are then projected back onto the input sequence, distributing the aggregated information.

Formally, we define a learnable query matrix $Q \in \mathbb{R}^{M \times C}$, where each row corresponds to a latent token. The key and value matrices, $K, V \in \mathbb{R}^{N \times C}$, are obtained by applying deep residual multi-layer perceptrons (MLPs) detailed in Appendix B to the input $X$. Compared to just a linear layer, these allow the model to learn higher-order feature interactions and deeper nonlinear transformations. Refer to Appendix G for ablation studies.

The matrices $Q$, $K$, and $V$ are first split along the feature dimension into $H$ heads, each of dimension $D = C/H$. Then, for encoding, each head performs SDPA with a scaling factor $s = 1$:

$$Z_h = \text{SDPA}(Q_h, K_h, V_h, s = 1). \tag{5}$$

Here, $Q_h \in \mathbb{R}^{M \times D}$, $K_h, V_h \in \mathbb{R}^{N \times D}$ are query, key, and value matrices belonging to head $h$ and $Z_h \in \mathbb{R}^{M \times D}$ is the latent sequence for head $h$. For decoding and propagating information back to the input tokens, we perform a second SDPA operation, swapping the roles of queries and keys and using the latent sequence as values:

$$Y_h = \text{SDPA}(K_h, Q_h, Z_h, s = 1) \tag{6}$$

where $Y_h \in \mathbb{R}^{N \times D}$ is the output for each head. Similar to multi-head self attention, the outputs from all heads are concatenated along the feature dimension and passed through a final linear projection to mix information across heads. As the query matrix has only $M$ tokens, the cost of SDPA calls in Eq. 5 and Eq. 6 is $\mathcal{O}(NM)$. PyTorch code for the multi-head implementation is presented in Figure 4. Note that we use a scaling factor $s = 1$ instead of $\sqrt{D}$ in typical transformers (Vaswani et al., 2017)) in SDPA. This modification is explained in Appendix G.

**Low-rank communication.** The two-step attention process can be written as

$$Y_h = (W_{\text{decode},h} \cdot W_{\text{encode},h}) \cdot V_h \tag{7}$$

where

$$W_{\text{encode},h} = \text{softmax}(Q_h \cdot K_h^T) \in \mathbb{R}^{M \times N}, \text{ and}$$
$$W_{\text{decode},h} = \text{softmax}(K_h \cdot Q_h^T) \in \mathbb{R}^{N \times M}. \tag{8}$$

Note that softmax is taken along the row-dimension. We define

$$W_h = W_{\text{decode},h} \cdot W_{\text{encode},h} \in \mathbb{R}^{N \times N} \tag{9}$$

as the dense global communication matrix with rank at most $M$. This low-rank structure, illustrated in Figure 2, enables efficient all-to-all communication without explicitly forming $W_h$; instead, $W_{\text{encode},h}$ and $W_{\text{decode},h}$ are applied sequentially, resulting in an overall cost of $\mathcal{O}(MN)$ per head.

**Discussion on the design principles of FLARE.** We defer to Appendix F a deeper analysis of how FLARE's latent tokens enable gather–scatter communication (acting as selective pooling hubs and broadcasters), why the symmetry between encoding and decoding operators promotes stable information flow, and how fixing latent queries highlights a tradeoff between query dynamics and the need for expressive key/value projections. Together, these perspectives clarify the structural principles that underlie FLARE's efficiency and expressivity.

Table 1: (Top) relative $L_2$ error ($\times 10^{-3}$) and (bottom) parameter count for different models across PDE benchmark problems. The best results (smallest error) are made bold, and the second best results are underlined. A backslash ($\backslash$) indicates that the model cannot be applied to the benchmark, and tilde ($\sim$) indicates that the model is prohibitively slow on the benchmark.

| Model | Elasticity | Darcy | Airfoil | Pipe | DrivAerML-40k | LPBF |
|---|---|---|---|---|---|---|
| Vanilla Transformer (Vaswani et al., 2017) | 5.37 660k | **4.38** 660k | 6.28 660k | $\sim$ | $\sim$ | $\sim$ |
| PerceiverIO (Jaegle et al., 2021a) | 28.0 1.87m | 20.6 1.87m | 7.65 1.87m | 6.90 1.87m | 248 1.87m | 23.1 1.87m |
| GNOT (Hao et al., 2023) | 13.3 4.87m | 16.9 4.90m | 103 4.90m | 5.89 4.90m | 115 4.87m | 24.3 4.87m |
| LNO (Wang & Wang, 2024a) | 9.25 1.83m | 7.64 762k | 17.8 762k | 8.10 762k | 146 762k | 24.7 762k |
| Transolver w/o conv (Wu et al., 2024) | 6.40 713k | 18.6 713k | 8.24 713k | 4.87 713k | 70.5 713k | 20.4 713k |
| Transolver with conv (Wu et al., 2024) | $\backslash$ | 5.94 2.8m | 5.50 2.8m | 3.90 2.8m | $\backslash$ | $\backslash$ |
| **FLARE (ours)** | **3.38** 592k | 5.10 691k | **4.28** 691k | **2.85** 625k | **60.8** 691k | **18.5** 625k |

**FLARE block.** Figure 2 (left) illustrates a single FLARE block. Given input tokens $X \in \mathbb{R}^{N \times C}$, the output of an FLARE block is computed as

$$X = X + \text{FLARE}\left(\text{LayerNorm}\left(X\right)\right)$$
$$X = X + \text{ResMLP}\left(\text{LayerNorm}\left(X\right)\right). \tag{10}$$

Here, ResMLP (He et al., 2016) denotes a residual MLP block detailed in Appendix B, and LayerNorm denotes layer normalization (Ba et al., 2016) operation. To summarize, a FLARE block consists of a token mixing operation via FLARE, a pointwise residual MLP, and layer normalization in pre-norm format (Xiong et al., 2020). Deep residual MLPs within the block enable complex, token-level feature transformations and improve accuracy.

**Overall design.** The overall architecture is given by $B$ sequential FLARE blocks sandwiched between an input projection and an output projection, which are detailed in Appendix B. Such a design enables the model to efficiently integrate local and global information across multiple layers, making it well suited for large-scale, high-dimensional data such as point clouds.

### 3.3 SPECTRAL ANALYSIS

The matrix $W_h$ in Eq. 9 represents the attention weights between $N$ tokens, where $[W_h]_{ij}$ quantifies how much token $j$ communicates to token $i$ within head $h$. Since $W_h$ has rank at most $M$, the model can capture at most $M$ independent global communication patterns per head. The eigenvalues of $W$ indicate the relative importance or energy of each latent dimension in forming the attention matrix $W$. We apply an algorithm to obtain the eigen decomposition of $W$ in $\mathcal{O}(M^3 + M^2 N)$ time compared to $\mathcal{O}(N^3)$ for a dense communication matrix. The algorithm is predicated on computing the eigenspectra of an $M \times M$ matrix $JJ^T$ where $J \in \mathbb{R}^{M \times N}$ is chosen such that $J^T J$ is similar to $W$. The algorithm is detailed in Section C.1, and summarized in Algorithm 1. Figure 13 presents the $M$ nonzero eigenvalues of $W_h$ for an FLARE model trained on the elasticity benchmark problem with 972 points per input. The distinct spectra of the heads indicates that each head learns distinct attention patterns.

The eigenvalue analysis detailed in Section C.2 shows that while FLARE provides capacity for rank-$M$ attention, the model learns to use only a small fraction of this in early blocks indicating effective compression. In deeper blocks, more of the latent capacity is utilized, with diverse spectral profiles across heads, validating our design choice of independent head-wise projections.

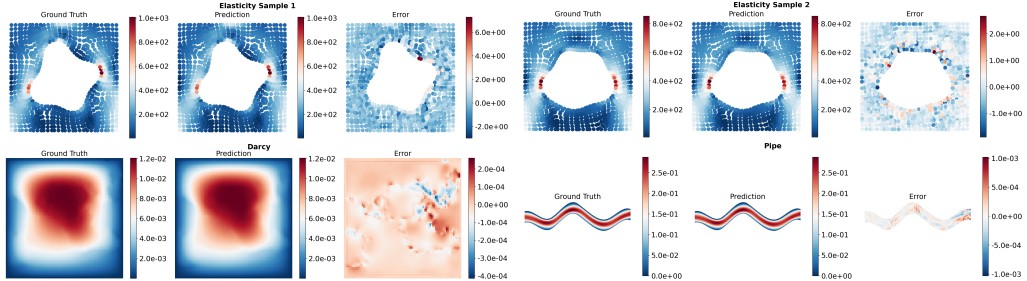

Figure 5: Qualitative results for FLARE on the Elasticity, Darcy and Pipe datasets. We show the ground truth, the model prediction, and the corresponding error (Ground Truth − Prediction).

# 4 BENCHMARK DATASET FOR ADDITIVE MANUFACTURING

We introduce a benchmark dataset of *laser powder bed fusion* (LPBF) simulations designed to evaluate surrogate models on large, irregular 3D geometries. LPBF is a widely used metal additive manufacturing process in which a laser fuses thin layers of powder, producing complex parts but often inducing *residual stresses* and *distortions* that can lead to build failures (Zhang et al.). To capture these effects, we simulate the thermo–mechanical LPBF process on thousands of geometries drawn from the Fusion 360 segmentation dataset (Lambourne et al., 2021).

Our benchmark task is *predicting the final vertical ($Z$) displacement field* for each geometry, a quantity directly associated with recoater-blade collisions and build-failure risk in LPBF. Each sample consists of the 3D mesh coordinates as input and the final $Z$-displacement at all nodes as output. The dataset spans a wide variety of part shapes, mesh resolutions, and deformation magnitudes, making it a challenging and practically meaningful testbed for large-scale surrogate modeling. Several visualizations are presented in Figure 14 and Figure 16. Additional simulation details, dataset statistics, and qualitative visualizations are provided in Appendix H.

# 5 EXPERIMENTS

## 5.1 PDE SURROGATE BENCHMARKS

**Benchmark problems.** We consider a diverse set of benchmark datasets (Table 3) for regressing PDE solutions on point clouds spanning structured and unstructured grids with up to 50,000 points. Note that FLARE is mesh-agnostic, and operates solely on the input point cloud. The 2D elasticity, darcy, airfoil, and pipe benchmarks (Li et al., 2020; 2023b) cover a wide range of physical phenomena, and the 3D DrivAerML benchmark (Ashton et al., 2024) provides automotive aerodynamic simulations. Visualizations of the elasticity, darcy, and pipe benchmark problems are presented in Figure 5. Additional details of the dataset are presented in Section D.2. We also introduce a 3D field-prediction benchmark derived from laser powder bed fusion (LPBF) simulations, with diverse 3D-printed parts containing up to 50,000 grid points (see Appendix H).

**Baselines.** We compare FLARE with state-of-the-art PDE surrogates: generic attention models (vanilla Transformer (Vaswani et al., 2017), PerceiverIO (Jaegle et al., 2021a)); attention-based PDE surrogates (Transolver (Wu et al., 2024), LNO (Wang & Wang, 2024a)); and the neural operator GNOT (Hao et al., 2023). We exclude graph-based models because graph connectivity is unavailable for most problems. Transolver++ (Luo et al., 2025) is a concurrent work; although we tested the recently released official implementation with reported hyperparameters, preliminary results did not match those reported in the paper, so we do not include it in our comparisons. We follow the experimental setup of Transolver as it is the preeminent surrogate model and attempt to match its parameter count. Note that Transolver can be instantiated in two configurations: *without convolution*, where point-to-point communication relies solely on physics attention, and *with convolution*, where convolution layers are added to inject information from neighboring points when the input grid is structured. We evaluate these two configurations separately to isolate the impact of convolution versus physics attention. In our model, we choose not to employ any convolution layers

Table 2: Accuracy (%) of different transformer models on Long Range Arena benchmark tasks Tay et al.. The best result (highest accuracy) is **bold** and the second best is underlined.

| Model | ListOps | Text | Retrieval | Image | Pathfinder-32 | Avg |
|---|---|---|---|---|---|---|
| Vanilla attention | **36.70** | 64.93 | 77.18 | 38.02 | 70.52 | 57.47 |
| Linear attention | 17.15 | **66.00** | 71.84 | 09.86 | **75.00** | 47.97 |
| Linformer | **36.70** | 53.00 | 64.72 | **41.88** | 70.09 | 53.28 |
| Norm attention | 17.10 | 63.08 | 76.07 | 36.94 | 70.15 | 52.67 |
| Performer | 35.90 | 64.21 | 68.42 | 35.36 | 53.83 | 51.54 |
| **FLARE (ours)** | 36.15 | 64.00 | **77.30** | 40.96 | 71.91 | **58.06** |

and rely entirely on FLARE for token mixing. Because these PDE problems are relatively small (up to 50,000 points), we train all models in FP32. As shown in Figure 8, the vanilla Transformer is drastically slower than FLARE and Transolver on large point clouds; accordingly, we evaluate it only on problems up to $\sim 10,000$ points.

**Discussion.** The results in Table 1 clearly demonstrate that the proposed FLARE architecture achieves the lowest relative $L_2$ error across all but one benchmark PDE problems, outperforming both LNO and Transolver on every dataset. Notably, FLARE also achieves these gains with a consistently lower parameter count than Transolver or LNO, highlighting its efficiency in addition to higher accuracy. These results underline the robustness and versatility of FLARE across diverse problem settings. We also note that the poor performance of Transolver without convolutions indicates that the inter-point communication via Transolver's built-in physics-attention mechanism is not enough. With convolutions, the input projections amass information from neighboring points, which in turn helps the physics attention learn the global structure.

Although the vanilla transformer is extremely effective on small-scale PDE problems, it becomes prohibitively slow on large point clouds due to its quadratic cost as illustrated in Figure 3. On the contrary, PerceiverIO (with only a single encoding and decoding step) performs poorly even with $M = 1,024$ latent tokens and $B = 8$ latent self-attention blocks. This validates our hypothesis that multiple latent self-attention operations can be unnecessary and potentially suboptimal so long as the projections are sufficiently expressive. This is because information loss during projection is not recoverable via latent self-attention alone. Instead, performing multiple (head-wise) parallel projections and reconstructions directly between the input and latent sequences preserves expressivity while simplifying the architecture.

## 5.2 Field-prediction on million-point geometries

Although the benchmark problems in Section 5.1 represent a wide variety of PDE problems, they are relatively small compared to industrial use cases that demand PDE solutions on complex geometries with millions of grid points (Ashton et al., 2024). So far, attention-based surrogate models have not been able to scale to million-scale regression problems due to quadratic time and memory complexity, as illustrated in Figure 3. The flash attention (Dao et al., 2022) algorithm has alleviated the memory bottleneck thanks to online softmax computation; however, these methods remain impractical due to their long training times. Furthermore, SOTA models such as Transolver and LNO cannot be implemented with off-the-shelf fused attention algorithms like flash attention because of the need to explicitly materialize the projection weights.

We demonstrate in Figure 6 that FLARE can scale to million-scale geometries by training on the DrivAerML dataset (Ashton et al., 2024) where each mesh is subsampled to contain one million points. These calculations are performed in mixed precision on a single Nvidia H100 80GB GPU provisioned through Google Cloud Platform. We note the clear trend in Figure 6 (left) that the error consistently decreases as we increase the number of FLARE blocks. To our knowledge, this is the first attention-based neural surrogate model trained on one million points on a single GPU without memory offloading or distributed computing.

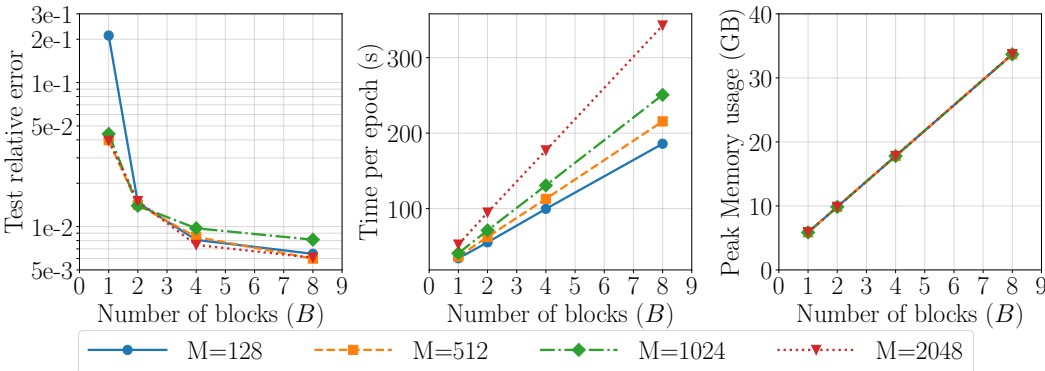

Figure 6: We train FLARE on the DrivAerML dataset (Ashton et al., 2024) with one million points per geometry on a single Nvidia H100 80GB GPU. We present (left) the test relative error, (middle) time per epoch (s), and (right) peak memory utilization (GB) as a function of the number of FLARE blocks ($B$) for different number of latent tokens ($M$).

### 5.3 LONG RANGE ARENA BENCHMARK PROBLEMS

To demonstrate that FLARE is not limited to PDE surrogate modeling, we also evaluate it on the Long Range Arena (LRA) benchmark suite (Tay et al.), which covers a diverse set of long-context tasks ranging from logical reasoning and text classification to retrieval, image classification, and visual pathfinding. While many top-performing LRA models rely on architectures specialized for 1D sequence structure (e.g., linear-attention kernels and state-space models such as MEGA (Ma et al., 2022), and Mamba (Gu & Dao)), FLARE is fully permutation-equivariant and makes no such structural assumptions, making its strong performance on LRA particularly noteworthy. As shown in Table 2, FLARE achieves the highest average accuracy across all LRA tasks, outperforming both general-purpose efficient-attention mechanisms (Linformer (Wang et al., 2020), Performer (Choromanski et al., 2020), Norm Attention (Qin et al., 2022), Linear Attention) and even vanilla self-attention on average. These results indicate that the low-rank formulation underlying FLARE provides a robust and broadly applicable inductive bias, extending beyond PDE surrogates to heterogeneous long-range reasoning tasks.

### 5.4 MODEL ANALYSIS AND ABLATIONS

In Appendix G, we scale FLARE to examine how expressivity trades off against time and memory complexity. A comprehensive set of complementary ablations is also provided in Appendix G, where we evaluate architectural choices such as the depth of residual MLPs in the key/value projections, the structure of the feedforward block, the effect of head dimension on accuracy, the impact of eliminating latent-space self-attention, and the role of independent versus shared latent tokens. Together, these experiments validate each component of the FLARE architecture and clarify how different design choices influence accuracy, efficiency, and scalability.

## 6 CONCLUSION

FLARE is a token mixing layer that bypasses the quadratic cost of self-attention by leveraging low-rankness. Mechanically, FLARE routes attention through a fixed-size latent sequence via cross-attention projection and unprojection. FLARE achieves SOTA accuracy on a set of diverse PDE benchmarks, and easily scales to PDE problems with million-scale geometries.

As transformers are the backbone of modern deep learning, we postulate that an efficient attention mechanism has several applications. We also identify potential areas for improvement: FLARE's reliance on deep residual MLPs can introduce sequential bottlenecks and increase latency, suggesting that further speedups are possible by addressing this issue. Additional enhancements include (1) incrementally increasing the number of latent tokens during training; (2) conditioning latent tokens on time for diffusion modeling; and (3) designing decoder-only variants for auto-regressive modeling.

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

Wentai Zhang, Brandon Abranovic, and Jacob Hanson-Regalado. Flaw detection in metal additive manufacturing using deep learned acoustic features.

## A    EXPLANATION OF CRITERIA IN FIGURE 1

Figure 1 compares PerceiverIO, Transolver, LNO, and FLARE across several architectural and computational criteria. Below, we provide detailed explanations for each heading used in the table.

**Asymptotic evaluation cost.**    This criterion measures the dominant computational complexity of processing a sequence of $N$ tokens for a model with $B$ blocks and $M$ latent tokens, ignoring feed-forward networks (which are always linear in $N$). PercieverIO incurs $MN$ encoding/decoding plus $BMN$ cross-attention cost; LNO incurs a one-time $MN$ encoding/decoding cost plus $BM^2$ latent self-attention cost. Transolver performs encoding and decoding *in every block*, leading to $2BMN + BM^2$ overall complexity. FLARE performs two cross-attention operations per block and eliminates latent self-attention entirely, achieving strictly linear $\mathcal{O}(BMN)$ complexity.

**Sequential encode/decode blocks.**    This heading indicates whether the model repeatedly projects inputs to a latent representation and decodes them back across multiple blocks. Transolver and FLARE apply encode–decode operations in every block, allowing global context to accumulate over depth. PerceiverIO and LNO apply only a single encode–decode step, limiting the model's ability to refine latent information over the full network depth.

**Headwise parallel encode/decode.**    This criterion describes whether each attention head independently projects the input into latent space. PerceiverIO and FLARE perform full per-head parallel encoding and decoding, yielding $H$ distinct projection matrices and enabling headwise specialization. Transolver ties projection weights across all heads, reducing diversity. LNO uses a single shared projection for all heads.

**Encoder/decoder coupling.**    Encoder/decoder coupling refers to whether the projection used to encode into the latent space is tied to the projection used to decode back to the input sequence. Transolver, LNO, and FLARE tie these weights, providing symmetry and parameter efficiency. PerceiverIO learns encoder and decoder weights independently.

**Latent space attention.**    This heading denotes whether the architecture performs explicit self-attention over the latent tokens. Latent self-attention increases expressivity but adds $M^2$ cost per block. PerceiverIO, LNO, and Transolver all perform latent self-attention. FLARE omits it entirely to preserve linear-time complexity and avoid additional quadratic dependence on $M$.

**Fused-attention compatibility.**    This criterion indicates whether the model's attention operations can be implemented using highly optimized fused SDPA kernels (e.g., FlashAttention). PerceiverIO and FLARE are fully compatible, enabling high GPU throughput. Transolver and LNO rely on feature-projection layers and explicit $M \times N$ operations that cannot be expressed as fused SDPA, limiting hardware efficiency at scale.

Together, these clarifications highlight the fundamental trade-offs between prior latent-attention architectures and motivate FLARE's design, which combines the depth and coupling benefits of Transolver with the linear complexity and hardware efficiency of cross-attention.

```
1  import torch.nn.functional as F
2  def flare_multihead_mixer_inefficient(Q, K, V):
3
4      # Args - Q: [H, M, D], K, V: [B, H, N, D]
5      # Ret - Y: [B, H, N, D]
6
7      # Compute projection weights
8      scores = Q @ K.mT                       # [B H M N]
9      W_encode = F.softmax(scores, dim=-1)    # [B H M N]
10     W_decode = F.softmax(scores.mT, dim=-1) # [B H N M]
11
12     # Encode: Project to latent sequence (M tokens)
13     Z = W_encode @ V
14
15     # Decode: Project back to input space (N tokens)
16     Y = W_decode @ Z
17
18     return Y
```

Figure 7: Pseudocode of FLARE if attention kernel is not available. See Figure 4 for efficient implementation.

## B  ARCHITECTURE DETAILS

### B.1  INPUT/ OUTPUT PROJECTION

**ResMLP.**  We implement a residual MLP block to serve as a flexible non-linear function approximator. Given input/output dimensions $C_i$ and $C_o$, the layer first applies a linear transformation to a hidden space of size $C_h$, followed by $L$ residual layers, each consisting of a linear layer with GELU activation (Hendrycks & Gimpel, 2016). These are the only instances of pointwise nonlinear activations in the model. An optional input residual connection is applied after the first layer when $C_i = C_h$, and an optional output residual connection is applied at the end when $C_h = C_o$. The final output is projected to dimension $C_o$ via a linear layer. This design allows control over depth and expressivity while preserving stability through residual connections.

**Input projection.**  The input projection consists of a ResMLP with $L = 2$, $C_i$ is the input feature dimension, and $C_h = C_o$ are set to $C$, the feature dimension of the model.

**Output projection.**  The output projection consists of a Layer Norm (Ba et al., 2016) followed by a ResMLP with $C_i = C$, $L = 2$, and $C_o$ is the output label dimension.

### B.2  FLARE BLOCK

The FLARE block illustrated in Figure 2, and detailed in Section 3, consists of the pointwise ResMLP layer, and the FLARE token mixer. For the ResMLP, we set $C_i = C_h = C_o = C$, set the number of layers to 3, and allow residuals to flow through the entire block.

**FLARE.**  FLARE consists of two ResMLPs for key/value projections, the token operation described in Figure 4, and an output projection. For the key/value projections, we set $C_i = C_h = C_o = C$, $L = 3$, and allow residuals to flow through the entire block. Figure 7 presents a mathematically equivalent PyTorch implementation for multi-head token-mixing operation without the fused SDPA kernels. The primary memory bottleneck in this implementation is materializing the $M \times N$ encoding weights and the $N \times M$ decoding weights. Its storage requirement is, thus, $\mathcal{O}(MN)$. Finally, the output projection is set to a single linear layer.

## C  SPECTRAL ANALYSIS

## C.1 EIGENANALYSIS PROCEDURE

We exploit the low-rank structure of the global communication matrix $W = W_{\text{decode}} \cdot W_{\text{encode}} \in \mathbb{R}^{N \times N}$ to obtain its eigen decomposition in $\mathcal{O}(M^3 + M^2 N)$ time, compared to the $\mathcal{O}(N^3)$ cost for a dense communication matrix. We find the eigen-decomposition of the FLARE attention matrix without actually forming the $N \times N$ matrix. We first note that $W_{\text{encode}}$ and $W_{\text{decode}}$ can be written in terms of the exponentiated score matrix $A = \exp(Q \cdot K^T) \in \mathbb{R}^{M \times N}$ as

$$W_{\text{encode}} = \Lambda_M \cdot A, \text{ and } W_{\text{decode}} = \Lambda_N \cdot A^T, \tag{11}$$

where $\Lambda^M \in \mathbb{R}^{M \times M}$, and $\Lambda^N \in \mathbb{R}^{N \times N}$ are diagonal matrices whose entries are

$$[\Lambda_M]_m = \frac{1}{\sum_{n=1}^N [A]_{m,n}}, \text{ and } [\Lambda_N]_n = \frac{1}{\sum_{m=1}^M [A]_{m,n}}. \tag{12}$$

Thus we have

$$W = \Lambda_N A^T \Lambda_M A \tag{13}$$

as the low-rank attention matrix. We observe that $W$ is similar to $J^T J \in \mathbb{R}^{N \times N}$ where $J = \Lambda_M^{1/2} A \Lambda_N^{1/2} \in \mathbb{R}^{M \times N}$. This is because

$$\begin{aligned} W = \Lambda_N A^T \Lambda_M A &= \underbrace{(\Lambda_N^{1/2} \Lambda_N^{1/2})}_{\Lambda_N} A^T \underbrace{(\Lambda_M^{1/2} \Lambda_M^{1/2})}_{\Lambda_M} A \underbrace{(\Lambda_N^{-1/2} \Lambda_N^{1/2})}_{I_N} \\ &= \Lambda_N^{1/2} \underbrace{(\Lambda_N^{1/2} A^T \Lambda_M^{1/2})}_{J^T} \underbrace{(\Lambda_M^{1/2} A \Lambda_N^{1/2})}_{J} \Lambda_N^{-1/2}. \end{aligned} \tag{14}$$

Thus $J^T J$ is symmetric, positive semi-definite, with rank at most $M$. Now suppose a singular value decomposition of $J$ as $J = U \Sigma V^T$ where $U \in \mathbb{R}^{M \times M}$ and $V \in \mathbb{R}^{N \times M}$ are the matrices whose columns are the left and right singular vectors of $J$ respectively, and $\Sigma \in \mathbb{R}^{M \times M}$ is the diagonal matrix of singular values. Then, we obtain

$$J^T J = V \Sigma \underbrace{U^T U}_{I_M} \Sigma V^T = V \Sigma^2 V^T, \tag{15}$$

and

$$W = \Lambda_N^{1/2} V \Sigma^2 V^T \Lambda_N^{-1/2}. \tag{16}$$

Post-multiplying both sides by $\Lambda_N^{1/2} V$, we have

$$W(\Lambda_N^{1/2} V) = (\Lambda_N^{1/2} V) \Sigma^2. \tag{17}$$

Therefore, the $M$ nonzero eigenvalues of $W$ are the squares of the singular values of $J$, and the corresponding eigenvectors are the columns of $\Lambda_N^{1/2} V$. Obtaining the eigenvalues and eigenvectors of $W$ this way requires the singular value decomposition of $J \in \mathbb{R}^{M \times N}$. We can do better by relating $V$ and $\Sigma$ to the eigen decomposition of $J J^T$. Consider the matrix $J J^T \in \mathbb{R}^{M \times M}$ with singular value decomposition, we have

$$J J^T = U \Sigma \underbrace{V^T V}_{I_M} \Sigma U^T = U \Sigma^2 U^T. \tag{18}$$

We note that the nonzero eigenvalues of $W$ are the same as the singular values of $J J^T$. To obtain the eigenvectors of $W$, we need an expression for $V$ in terms of $U$, $J$, and $\Sigma$. We do so by noting that

$$J^T U = (V \Sigma U^T) U = V \Sigma \implies V = J^T U \Sigma^{-1}. \tag{19}$$

Therefore, the eigenvectors of $W$ are

$$\Lambda_N^{1/2} V = \Lambda_N^{1/2} J^T U \Sigma^{-1}. \tag{20}$$

To find the eigenvalues and eigenvectors of $W$, one only needs to compute the eigen-decomposition of the $M \times M$ matrix $J^T J$. The overall algorithm, summarized in Algorithm 1, takes $\mathcal{O}(M^3 + NM^2)$ time where the $\mathcal{O}(M^3)$ is for computing the SVD of $J J^T$.

---

**Algorithm 1** Eigenvalues and Eigenvectors from $Q$, $K$

---

**Require:** $Q \in \mathbb{R}^{M \times D}$, $K \in \mathbb{R}^{N \times D}$
1: $A \leftarrow \exp\left(Q \cdot K^T\right)$
2: $L_N \leftarrow \text{diag}(1/\sum_{m=1}^{M}[A]_{m,n})$
3: $L_M \leftarrow \text{diag}(1/\sum_{n=1}^{N}[A]_{m,n})$
4: $J \leftarrow L_M^{1/2} \cdot A \cdot L_N^{1/2}$
5: Compute SVD: $U\Sigma^2 U^T \leftarrow JJ^T \in \mathbb{R}^{M \times M}$
6: Eigenvalues $\leftarrow \Sigma^2$
7: Eigenvectors $\leftarrow L_N^{1/2} J^T U \Sigma^{-1}$
8: **return** Eigenvalues, Eigenvectors

---

## C.2 Qualitative Analysis

The matrix $JJ^T$, being $M \times M$, captures the structure of this latent space and provides insights into how these $M$ dimensions contribute to the attention mechanism. Large eigenvalues correspond to dominant latent dimensions that contribute significantly to attention patterns. If some eigenvalues are small or zero, those latent dimensions contribute little, suggesting redundancy in the latent space. The number of nonzero eigenvalues gives the effective rank of $W$, which reflects how many independent patterns the attention mechanism captures.

Figure 13 (middle) presents the $M = 64$ nonzero eigenvalue spectra of an FLARE model trained on the elasticity dataset with $M = 64$ latents. Some observations are as follows. In all blocks, especially block 1, the eigenvalues drop sharply within the first $10 - 20$ indices. This indicates that even though the communication matrices $W_h$ could have rank up to $M = 64$, most of the energy (information) is captured by a much smaller subset of modes. This result validates the hypothesis that the global communication pattern is inherently low-rank.

We also observe that the eigenvalue curves in block 5 and block 8 decay more slowly, retaining more moderate-magnitude eigenvalues beyond index 20–30. This indicates that as depth increases, the attention mechanism seems to leverage more of the latent space — i.e., the effective rank increases. This shows that deeper layers learn richer global dependencies, and the model may be using more of the projection capacity in later blocks.

Finally, we note that the curves for different heads (colors) have distinct decay patterns, especially in later blocks. This reinforces the claim that separate projection matrices per head enables specialized routing. This supports the idea that FLARE benefits from head-wise diversity, rather than using shared projections like in Transolver (Wu et al., 2024).

# D Benchmarking and Comparison

## D.1 Benchmark metrics

The primary evaluation metric for all benchmarks is the relative $\mathcal{L}_2$ error, which quantifies the normalized discrepancy between the predicted solution $\hat{u}$ and the ground truth solution $u$ over all points in the domain. For a given test sample, the relative $L_2$ error is defined as:

$$\text{Relative } L_2 = \frac{\|\hat{u} - u\|_2}{\|u\|_2} \tag{21}$$

where $\|\cdot\|_2$ denotes the standard Euclidean norm. For datasets where each sample consists of $N$ points (or grid locations), this expands to:

$$\text{Relative } L_2 = \frac{\left(\sum_{i=1}^{N}(\hat{u}_i - u_i)^2\right)^{1/2}}{\left(\sum_{i=1}^{N} u_i^2\right)^{1/2}}. \tag{22}$$

The reported metric is averaged over all test samples.

Table 3: Summary of PDE benchmarks.

| Benchmark | #Dim | Grid Type | #Points | #Input/ Output Features | #Train/ Test Cases |
|---|---|---|---|---|---|
| Elasticity | 2D | Unstructured | 972 | 2 / 1 | 1000 / 200 |
| Darcy Airfoil Pipe | 2D | Structured | 7,225 11,271 16,641 | 2 / 1 | 1000 / 200 |
| DrivAerML-40k LPBF | 3D | Unstructured | 40,000 1,000–50,000 | 3 / 1 | 387 / 97 1100 / 290 |

## D.2 BENCHMARK DATASETS

We evaluate all models on five benchmark datasets and our proposed AM dataset, each designed to assess generalization, scalability, and robustness to domain irregularity in PDE surrogate modeling. A summary is provided in Table 3.

**Elasticity.** This benchmark estimates the inner stress distribution of elastic materials based on their structure. Each sample consists of a tensor of shape $972 \times 2$ representing the 2D coordinates of discretized points, and the output is the stress at each point ($972 \times 1$). The dataset contains 1,000 training and 200 test samples with different material structures (Li et al., 2023b).

**Darcy.** This benchmark models fluid flow through a porous medium. The porous structure is discretized on a $421 \times 421$ regular grid, downsampled to $85 \times 85$ for main experiments. The output is the fluid pressure at each grid point. There are 1,000 training and 200 test samples with varying medium structures (Li et al., 2021).

**Airfoil.** This task estimates the Mach number distribution around airfoil shapes. The input geometry is discretized into a structured mesh of shape $221 \times 51$, representing deformations of the NACA-0012 profile, and the output is the Mach number at each mesh point. The dataset includes 1,000 training and 200 test samples with unique airfoil designs (Li et al., 2023b).

**Pipe.** This benchmark predicts horizontal fluid velocity in pipes with varying internal geometries. Each sample is represented as a $129 \times 129 \times 2$ tensor encoding mesh point positions, with the output being the velocity at each mesh point ($129 \times 129 \times 1$). The dataset consists of 1,000 training and 200 test samples, with pipe shapes generated by controlling the centerline (Li et al., 2023b).

**DrivAerML-40k.** The DrivAerML dataset (Ashton et al., 2024) has high-fidelity automotive aero-dynamic simulations, featuring 500 parametrically morphed DrivAer vehicles. CFD simulations are performed on 160 million volumetric mesh grids using hybrid RANS-LES, the highest-fidelity scale-resolving CFD approach routinely deployed by the automotive industry (Ashton et al., 2024). Each sample in the dataset includes a surface mesh with approximately 8.8 million points and corresponding pressure values. Since no official dataset split is provided, an 80/20 random split is used, with 40,000 points sampled per case for training and evaluation.

**LPBF.** We introduce the laser powder bed fusion (LPBF) additive manufacturing dataset which involves field prediction on complex geometries. In metal additive manufacturing, variations in design geometry can affect the dimensional accuracy of the part and lead to shape distortions. We perform several LPBF process simulations of a set of geometries to obtain the deformation field over the geometry. We select a subsample of the dataset with up to 50,000 points per geometry and divide it into 1,100 training and 290 cases. We train models to learn the $Z$ (vertical) component of the deformation field at each grid point. Additional details are provided in Appendix H.

Table 4: Standard training configuration on PDE datasets. Identical values are grouped for clarity.

| Dataset | Batch Size | Weight Decay | Learning Rate | Epochs | Loss |
|---|---|---|---|---|---|
| Elasticity | 2 | $10^{-5}$ | $10^{-3}$ | 500 | Rel. $L_2$ |
| Darcy | | | | | Rel. $L_2 + 0.1\, L_g$ |
| Airfoil | 2 | $10^{-5}$ | $10^{-3}$ | 500 | Rel. $L_2$ |
| Pipe | | | | | Rel. $L_2$ |
| DrivAerML | 1 | $10^{-2}$ | $10^{-3}$ | 500 | Rel. $L_2$ |
| LPBF | | $10^{-4}$ | | 250 | |

Note: For Darcy test case, we include an additional gradient regularization term $L_g$ following Transolver (Wu et al., 2024).

Table 5: Model configurations for FLARE for PDE datasets. Identical values are grouped for clarity.

| Dataset | #Heads ($H$) | #Latents ($M$) | #Blocks ($B$) | #Features ($C$) |
|---|---|---|---|---|
| Elasticity | 8 | 64 | 8 | 64 |
| Darcy | 16 | 256 | | |
| Airfoil | 8 | 256 | 8 | 64 |
| Pipe | 8 | 128 | | |
| DrivAerML-40k | 8 | 256 | 8 | 64 |
| LPBF | 16 | 256 | | |

### D.3 BENCHMARK MODELS AND TRAINING DETAILS

We follow the recommended hyperparameter configuration for LNO, Transolver, and GNOT wherever possible. For consistency, we set the number of blocks to $B = 8$, and strive to match the parameter counts of Transolver without (w/o) convolution for all models. As such, the hidden feature dimension for vanilla transformer, is set to $C = 80$. Its head dimension is set to $D = 16$ and MLP ratio is 4 which is typical for transformers (Vaswani et al., 2017). For FLARE, we set the hidden feature dimension to $C = 64$, and use $H = 8$ or 16 heads, which result in a head dimension of $D = 8$ or 4 respectively. The number of latents is chosen from $M \in \{64, 128, 256\}$ depending on the problem. As PerceiverIO was not designed to be a surrogate model, we generously set its channel dimension to $C = 128$, and number of latents to $M = 1,024$. Furthermore, the input and output projections for vanilla transformer, perceiver, and FLARE are held consistent to facilitate an equitable comparison of their point-to-point communication schemes.

We evaluate Transolver, GNOT with the hyperparameter configurations provided in Wu et al. (2024) and LNO on the ones provided in Wang & Wang (2024a) on the 2D test cases. For the remaining problems, we choose the best performing parameter set from the 2D cases.

Unless otherwise stated, all models are trained with the AdamW (Loshchilov & Hutter, 2019) optimizer ($\beta_1 = 0.9$, $\beta_2 = 0.999$) to minimize the relative $L_2$ error. We use the OneCycleLR (Smith & Topin, 2019) scheduler with 10% of epochs dedicated to warming-up to a learning rate of $10^{-3}$, followed by cosine decay. We train on LPBF for 250 epochs, and 500 epochs for all other models. Note that we use gradient clipping with max_norm= 1.0 unless otherwise stated. Unless otherwise stated, the weight decay regularization parameter is set to $10^{-5}$ for the 2D test cases, $10^{-4}$ for DrivAerML-40k, and $10^{-4}$ for LPBF. The batch size is set to 2 for the 2D problems and 1 for the 3D problems unless otherwise stated. Unless otherwise stated, all models are trained in full precision (FP32).

**Vanilla transformer.** For the vanilla transformer, we set the hidden feature dimension to $C = 80$, and choose $H = 5$ heads so that the head dimension $D = 16$. The number of blocks is set to $B = 8$, and the MLP ratio for the feedforward block is set to 4. The vanilla transformer can be prohibitively slow for test cases with over 10,000 test points.

**PerceiverIO.**   For PerceiverIO, we use $B = 8$, $C = 128$, $H = 8$, and set the latent sequence length to $M = 512$ for all test cases.

**Transolver.**   Transolver (Wu et al. (2024)) introduces Physics-Attention: each mesh point is softly assigned to a few learnable *slices*, shrinking thousands of points to only tens of tokens. Self-attention runs on this compact set and the tokens are then desliced back to points. Following the hyperparameter recommendations in the code of Wu et al. (2024), Transolver is trained with 30% of the steps dedicated to warm-up, and gradient clipping with max_norm = 0.1. The recommended batch size for Transolver is 1 for elasticity, and 4 for the remaining 2D problems. For the 3D problems, we set the batch size to 1.

**Latent Neural Operator (LNO).**   LNO (Wang & Wang (2024b)) moves computation into a small latent space. An embedder lifts the input field, cross-attention compresses $N$ points into $M$ latent tokens, transformers act solely on these tokens, and a decoder maps them to any query location. In line with the recommended hyperparameter configuration in Wang & Wang (2024b), we set $\beta_2 = 0.99$ in the AdamW optimizer, do warmup for the first 20% of epochs, and clip gradient norms greater than 1,000. The number of hidden features are set to $C = 192$ for elasticity, and $C = 128$ for all other test cases. The number of residual layers is set to 3 for elasticity and 4 for other test cases. The number of latent self attention blocks is 8 for pipe and airfoil test cases and 4 for all other cases. The number of latent modes is set to 256 for all test cases. The batch size during training is set to 4 for the 2D test cases and 1 for the 3D test cases. The LNO code also recommends weight decay regularization of $5 \cdot 10^{-5}$ for the 2D test cases.

In running our experiments, we noticed discrepancies between our LNO results and those presented in their article (Wang & Wang, 2024a). Upon further investigation, we found that the datasets used in the LNO paper and in the original Transolver paper are not the same. For example, in the Elasticity dataset, LNO was trained and tested on a $1000/1000$ split, whereas Transolver used a $1000/200$ split. In the Darcy dataset, LNO employed a higher resolution of $241 \times 241$, compared to $85 \times 85$ in Transolver. Because these differences make direct comparison unreliable, all models (including LNO and Transolver) were re-trained and evaluated on the standardized Transolver splits and resolutions to ensure fairness.

**General Neural Operator Transformer (GNOT).**   GNOT (Hao et al. (2023)) employs heterogeneous normalized attention, separately normalizing keys and values, to fuse multiple input fields on irregular meshes. A learnable geometric gate decomposes the domain and routes tokens to scale-specific expert MLPs. Linear cost attention plus this gating scales to large problems and surpasses earlier operator learners.

Following the hyperparameter recommendations outlined in the code of Wu et al. (2024), GNOT is trained with 30% of the steps dedicated to warm-up, and gradient clipping with max_norm = 0.1. The recommended batch size for Transolver is 1 for elasticity, and 4 for the remaining 2D problems. The batch size is set to 2 for elasticity, 4 for the remaining 2D benchmarks and 1 for the 3D benchmarks.

**FLARE.**   For all problems, we employ $B = 8$ blocks with a feature dimension of $C = 64$. We set the number of residual layers and the number of key/value projection layers to 3, and vary the head dimension as $D \in \{4, 8\}$, the number of latent tokens as $M \in \{64, 128, 256\}$ The hyperparameters for each test case are presented in Table 5.

# E   FIELD-PREDICTION ON MILLION-POINT GEOMETRIES

All FLARE models in this study are trained in mixed precision to take advantage of the flash attention algorithm. We train for 500 epochs with the OneCycleLR scheduler (Smith & Topin, 2019) where the first $5\%$ of epochs are spent warming up to a learning rate of $5 \cdot 10^{-4}$ followed by cosine decay.

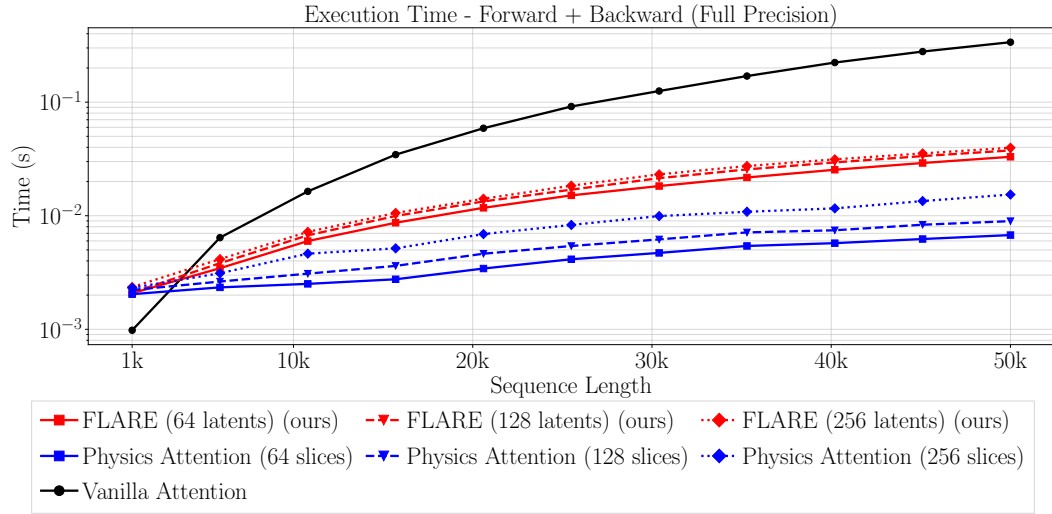

Figure 8: Execution times in FP32 for a single vanilla self-attention layer, a physics attention layer, and FLARE. The models are set to have approximately the same number of parameters as in Section 5.1. This calculation is performed on a single H100 80GB GPU. Note that the curves for FLARE are somewhat overlapping.

## F  DISCUSSION ON THE DESIGN PRINCIPLES OF FLARE.

**Latent tokens enable gather-scatter communication.**  In FLARE, information flows through latent tokens by first *gathering* from the input sequence and then *scattering* back. The encoding step can be understood as a gather (all-reduce) operation, where each latent token pools information from the input according to its learned query pattern. Formally, for latent query $q_m$,

$$z_m = \sum_{n=1}^{N} \frac{\exp(q_m \cdot k_n)}{\sum_{n'=1}^{N} \exp(q_m \cdot k_{n'})} \, v_n, \quad m = 1, \ldots, M, \tag{23}$$

$z_m$ aggregates input values $v_n$ with convex weights. When the similarity scores are sharp, $z_m$ emphasizes a few dominant inputs; when they are flatter, $z_m$ acts like a specialized *pooling token* that averages a select set of tokens. Across $M$ latents, this yields a compact set of global descriptors, each specializing in pooling different aspects of the input.

The decoding step is the dual scatter (broadcast) operation, but importantly, the broadcast is *selective*: each latent $z_m$ contributes only to the input tokens that assigned it high weight in the encoding step. Concretely,

$$y_n = \sum_{m=1}^{M} \frac{\exp(k_n \cdot q_m)}{\sum_{m'=1}^{M} \exp(k_n \cdot q_{m'})} \, z_m, \quad n = 1, \ldots, N, \tag{24}$$

the output token $y_n$ only receives substantial information from the latents whose query pattern matches its key $k_n$ strongly. In this sense, each latent acts as both a selective *pooling hub* and *broadcaster*, routing information back along the pathways that originally recruited it. Together, these gather–scatter stages form a low-rank, butterfly-like transformation: first contracting $N$ tokens into $M \ll N$ global features, and then expanding back to $N$ outputs, enabling efficient, yet expressive, global communication.

**Symmetry in latent token communication.**  The compression–expansion attention structure in FLARE is most effective when the encoding and decoding operators are structurally aligned. Specifically, $W_{\text{encode},h} = \text{softmax}(Q_h \cdot K_h^T)$ and $W_{\text{decode},h} = \text{softmax}(K_h \cdot Q_h^T)$ are transposes of each other up to diagonal scaling. We experimented with breaking this symmetry by using distinct query–key pairs for encoding and decoding, but observed no accuracy gains. This suggests that the near-adjoint relationship between $W_{\text{encode}}$ and $W_{\text{decode}}$, both derived from the same parameters,

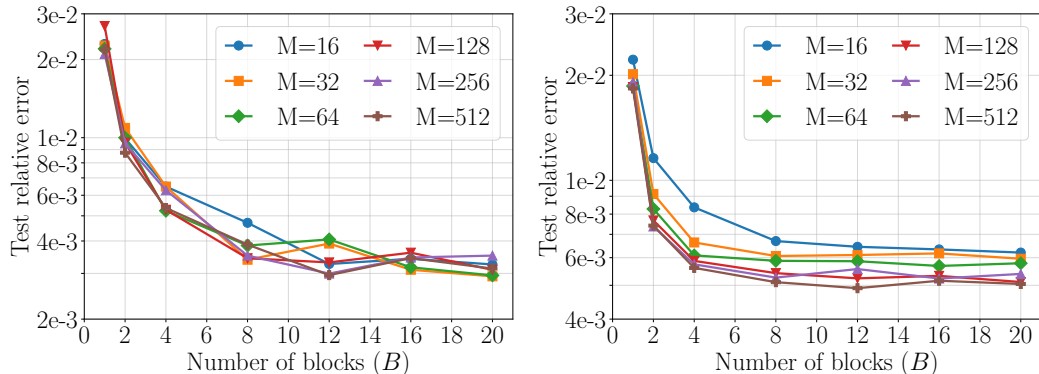

Figure 9: Effect of number of blocks ($B$) and number of latent tokens ($M$) on test accuracy on the elasticity (left) and darcy (right) test cases. Experiment details are presented in Appendix G.

may provide a form of mathematical optimality, ensuring stable information flow through latent tokens while reducing representational redundancy.

**Tradeoff between query dynamics and key/value expressivity.**    An important design choice in FLARE is to use fixed, input-independent queries $Q$, which constrain the flexibility of the attention pattern. Detaching $Q$ from $X$ allows for the low-rank communication structure, but requires compensating expressivity in the key and value projections. In practice, we find that deeper residual MLPs for $K$ and $V$ are crucial for capturing rich, feature interactions under this constraint. Conversely, one could imagine making $Q$ dynamic by conditioning it on $X$, which would shift more of the modeling burden to the query side and potentially allow shallower $K, V$ projections. Thus, FLARE embodies a clear tradeoff: fixing $Q$ encourages stability and efficiency, but places greater importance on the depth and expressivity of the key/value encoders.

## G    MODEL ANALYSIS AND ABLATIONS

**Time and memory complexity.**    Figure 3 illustrates the time and memory complexity of a single forward and backward pass for different attention schemes on long sequences. The experiment is done in mixed precision (FP16 in forward pass, FP32 in backward pass) using PyTorch's autocast functionality with $C = 128$ features and $H = 8$ heads for all models. The flash attention backend (Dao et al., 2022) is employed for SDPA wherever possible.

Although vanilla self-attention has the lowest memory cost thanks to the flash-attention algorithm, which eliminates the need to materialize the score matrices ($Q_h \cdot K_h^T$), its compute time still scales poorly with the sequence length. In contrast, the compute time for FLARE exhibits strong scaling with sequence length. Its memory requirement is marginally greater than vanilla attention due to the presence of deep residual networks for key/value projections, and due to the need to materialize $Z_h$, the latent sequence of $M$ tokens. As these costs are marginal compared to the SDPA operation, the curves for different $M$ values of FLARE are somewhat overlapping. Finally, the compute time for Physics Attention of Transolver (Wu et al., 2024) exhibits somewhat good scaling. However, its memory cost and compute time blow up for large slice counts due to the need for materializing the projection matrices.

**Number of blocks ($B$) and latent tokens ($M$).**    Figure 9 presents the test relative error of FLARE on the elasticity (left) and darcy (right) benchmark datasets as a function of the number of blocks ($B$) and the number of latent tokens ($M$). Figure 6 (left) presents the same for the DrivAerML dataset with one million points per geometry. In all cases, we note the favorable trend that relative error consistently decreases as we increase the number of blocks. Similarly, we observe that the relative error generally decreases with $M$, though the trend is not strictly monotonic. In the elasticity problem, improvements with rank diminish rapidly, indicating that global communication in that problem is fundamentally low-rank. On the other hand, increasing $M$ monotonically increases

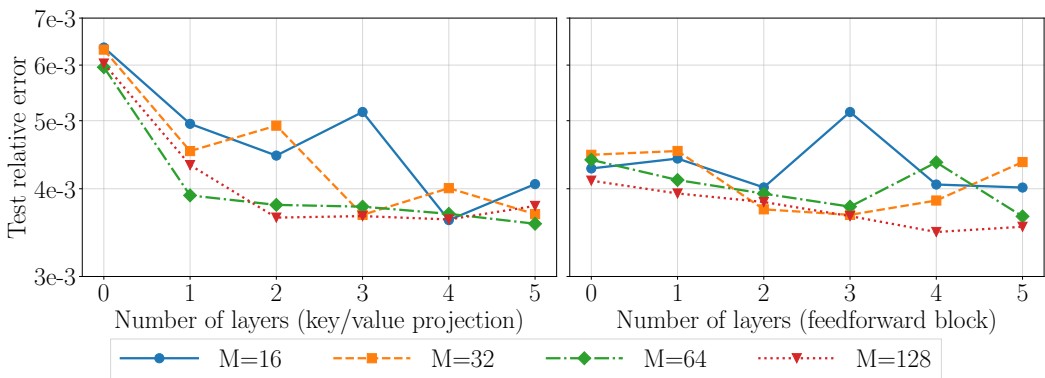

Figure 10: (Left) effect of the number of residual layers in key/ value projection, and (right) of residual layers in residual block on test accuracy. In both cases, deeper networks lead to greater accuracy.

performance on the darcy problem, indicating that the problem is *rank-limited*. This also explains why vanilla transformer with a full-rank attention pattern outperforms rank-deficient FLARE on the darcy problem. However, the accuracy gain comes at the cost of greater latency as the vanilla transformer is $\sim 5\times$ slower than FLARE on the Darcy problem. Figure 6 indicates that time per epoch (middle) and memory (right) scaling of FLARE with $B$ and $M$. Here, increasing $M$ leads to increased latency, and that increasing $M$ does not come at the cost of greater memory requirements.

**ResMLP depth: key/value projections.** A substantial distinction between vanilla self-attention and FLARE is the introduction of deep residual blocks for key and value projections in place of simple linear layers. In standard self-attention, queries ($Q_h$) and keys ($K_h$) determine the global communication pattern through $W_h = \text{softmax}(Q_h K_h^T / s)$, while values ($V_h$) carry the information to be communicated. All three are typically computed as shallow linear projections of the input $X$. In contrast, FLARE computes the attention pattern as $W_h = \text{softmax}(K_h Q_h^T) \cdot \text{softmax}(Q_h K_h^T)$, where the query embeddings $Q_h$ are learned parameters independent of the input. This makes the attention pattern less dynamic, motivating architectural modifications to enhance flexibility.

To address this, we replace the linear key and value projections with deep residual MLPs. Using residual networks for key and value encodings allows each token to learn richer and more structured features rather than shallow embeddings, which is particularly crucial in FLARE since the queries are fixed and cannot adapt to the input. Figure 10 (left) shows the impact of varying the number of residual layers in key/value projections and within the residual block on test accuracy for the elasticity benchmark dataset. We suspect that deeper key/value encodings lead to more meaningful and focused attention, encoding structured inductive priors beneficial to downstream prediction.

**ResMLP depth: feedforward block.** A second difference between the standard attention block and FLARE block is that we replace the feed-forward block in vanilla self-attention with a deep residual MLP. Preliminary experiments using standard feed-forward blocks led to training instabilities and poor convergence. In contrast, residual MLPs consistently enabled stable training and allowed us to increase model capacity. Figure 10 (right) indicates that increasing the number of residual layers leads to slight improvements in accuracy. Based on these results, we use three residual layers in both key/value projections and the residual block, as this provides a good trade-off between model capacity and computational cost in all subsequent experiments.

**Effect of head dimension and parallel low-rank projections.** A core hypothesis underlying FLARE is that each attention head implements an independent rank–$\leq M$ projection–reconstruction pathway. When multiple such pathways operate in parallel, the resulting attention operator becomes a mixture of low-rank factors, each capturing a distinct structural component of the underlying communication pattern. This is in contrast to Transolver, which shares projection weights across all heads, and LNO, which uses a single global projection: both designs restrict the model to learning

at most one or a few shared low-rank directions. FLARE, by contrast, allows each head to specialize in complementary routing patterns by assigning it an independent slice of the latent tokens.

To examine this hypothesis quantitatively, we vary the number of heads $H$ while keeping the total feature dimension $C = 64$ and number of blocks $B = 8$ fixed, thereby trading off head dimension $D = C/H$ against the number of parallel low-rank projections available to the model. As shown in Figure 11, FLARE consistently achieves the best accuracy for $D = 4$ or $D = 8$, outperforming configurations with larger head dimensions. This behavior is the reverse of what is commonly observed in standard transformers—where typical head dimensions are $D = 16$–32, but is entirely consistent with FLARE's architectural role for each head. Larger $D$ increases the per-head representational capacity but reduces the number of parallel low-rank factors; smaller $D$ yields more distinct heads, and thereby more distinct projection–reconstruction pathways, which better approximate a full attention map.

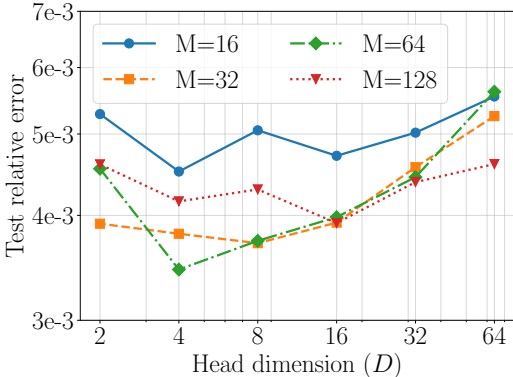

Figure 11: Effect of head dimension ($D$) on test accuracy. We design FLARE to work optimally for $D = 4$–8.

We additionally note a practical implication of this analysis: because the dot-product magnitudes are naturally small for $D \in \{4, 8\}$, we use a scaling factor of 1 rather than the usual $1/\sqrt{D}$ in these cases, simplifying implementation without sacrificing stability. Overall, the head-dimension ablation strongly supports our main architectural claim: FLARE benefits from *multiple parallel low-rank projections*, and enabling per-head independence is crucial for capturing diverse communication patterns.

**Ablation on latent-space blocks vs. FLARE blocks.** To directly investigate whether FLARE's encode–decode mechanism is responsible for the observed performance gains, we conduct a controlled ablation varying the number of latent-space self-attention blocks ($L_B$) and the number of full FLARE blocks ($B$). This experiment spans the continuum between a Perceiver/LNO-like architecture (few encode–decode operations and many latent-space blocks; bottom-left of Figure 12) and a FLARE-like architecture (many encode–decode blocks and no latent-space self-attention; top-right).

Across all parameter budgets, we observe a consistent trend: *increasing the number of latent-space blocks degrades accuracy while increasing computational cost*. Latent-space self-attention contributes additional parameters and a $\mathcal{O}(M^2)$ cost per block, yet does not improve the model's ability to capture global structure. In contrast, allocating the same compute to additional encode–decode blocks—which perform low-rank global mixing via cross-attention—steadily improves accuracy. The lowest errors in the entire grid occur in the top-right corner, corresponding to *zero* latent-space blocks and the largest number of FLARE blocks.

This ablation provides direct causal evidence supporting FLARE's architectural choice: global communication is most effectively learned by repeating the low-rank projection/unprojection pathway, not by performing nonlinear refinements inside the latent space. If one interprets the Perceiver ↔ FLARE spectrum as trading latent refinement for repeated global mixing, our results show that the optimal regime is decisively FLARE-like. For a fixed parameter budget and wall-clock time, the best strategy is to *reduce or eliminate latent-space attention and increase the number of encode–decode blocks*, validating the design principle that token mixing should occur through repeated low-rank attention projections rather than deeper latent-space transformers.

**Ablation on shared vs. independent per-head latent tokens.** To assess the effect of head-wise latent diversity—central to FLARE's design, we compare two variants: (i) *shared-latent* models, where all heads use the same latent sequence, and (ii) *independent-latent* models, where each head receives its own slice of the latent tokens. This isolates whether FLARE's expressivity arises merely

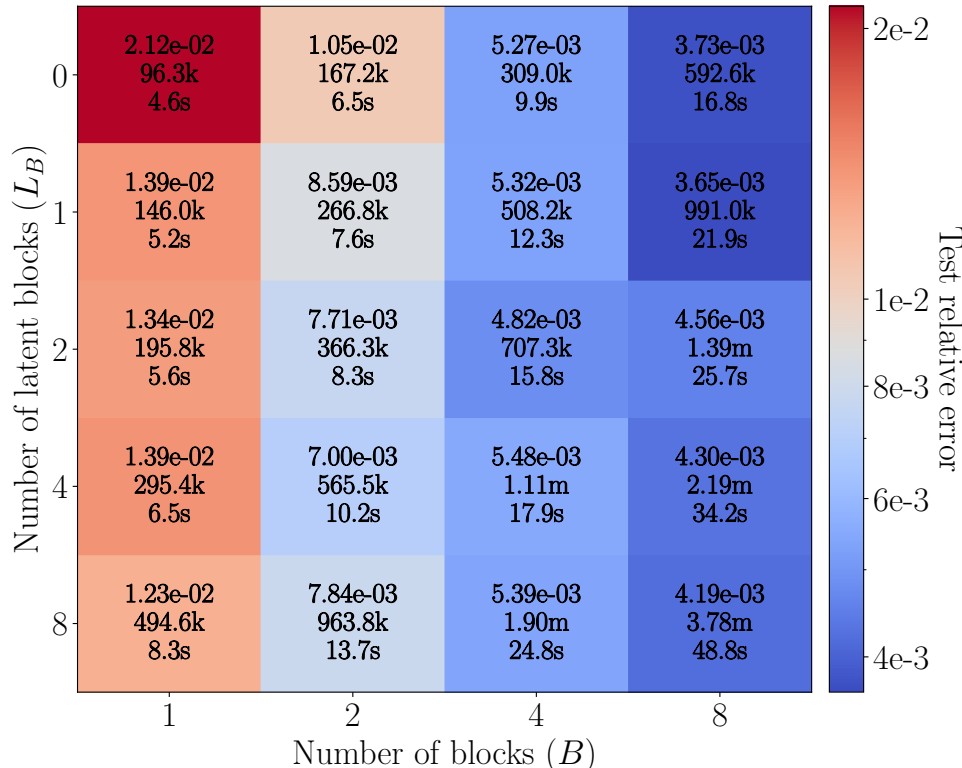

Figure 12: **Ablation on the number of latent-space self-attention blocks ($L_B$) versus the number of FLARE encode–decode blocks ($B$).** Each cell reports: (top) test relative error, (middle) parameter count, and (bottom) training time per epoch. The total cost of a model with $B$ FLARE blocks and $L_B$ latent blocks is $\mathcal{O}(B\,NM + BL_B\,M^2)$. The results show that increasing the number of latent-space blocks—as done in LNO-style models—yields worse accuracy and poorer speed/-parameter tradeoffs than allocating compute to additional FLARE blocks. The optimal regime lies near the *top-right corner*: many encode–decode blocks and *zero* latent-space blocks.

from the low-rank encode–decode structure or whether the ability of different heads to learn distinct low-rank projections provides additional modeling capacity.

The spectral plots in Figure 13 reveal a clear difference between the two settings. With shared latents, all heads exhibit nearly identical eigenvalue decay, implying that they compress and propagate information in similar ways. In contrast, independent latents produce noticeably different spectra across heads, particularly in deeper blocks, indicating that different heads discover complementary low-rank subspaces of the token-to-token communication structure. This diversity is modest in early layers but becomes increasingly pronounced as depth increases.

Crucially, the quantitative results mirror this qualitative behavior: across all depths, models with independent per-head latents achieve lower test relative error despite having comparable parameter counts. This provides causal evidence—not just descriptive visualization—that head-wise independence is beneficial. It supports our claim that FLARE's mixture of head-specific low-rank projections yields richer communication pathways than architectures that use a single shared projection (e.g., Transolver) or a single latent transformer (e.g., LNO, PerceiverIO). Finally, we note that applying the same spectral analysis to Transolver or LNO is not directly meaningful, since their latent-space transformations are nonlinear and not expressible as a single linear low-rank operator; FLARE's structure uniquely enables such eigenanalysis.

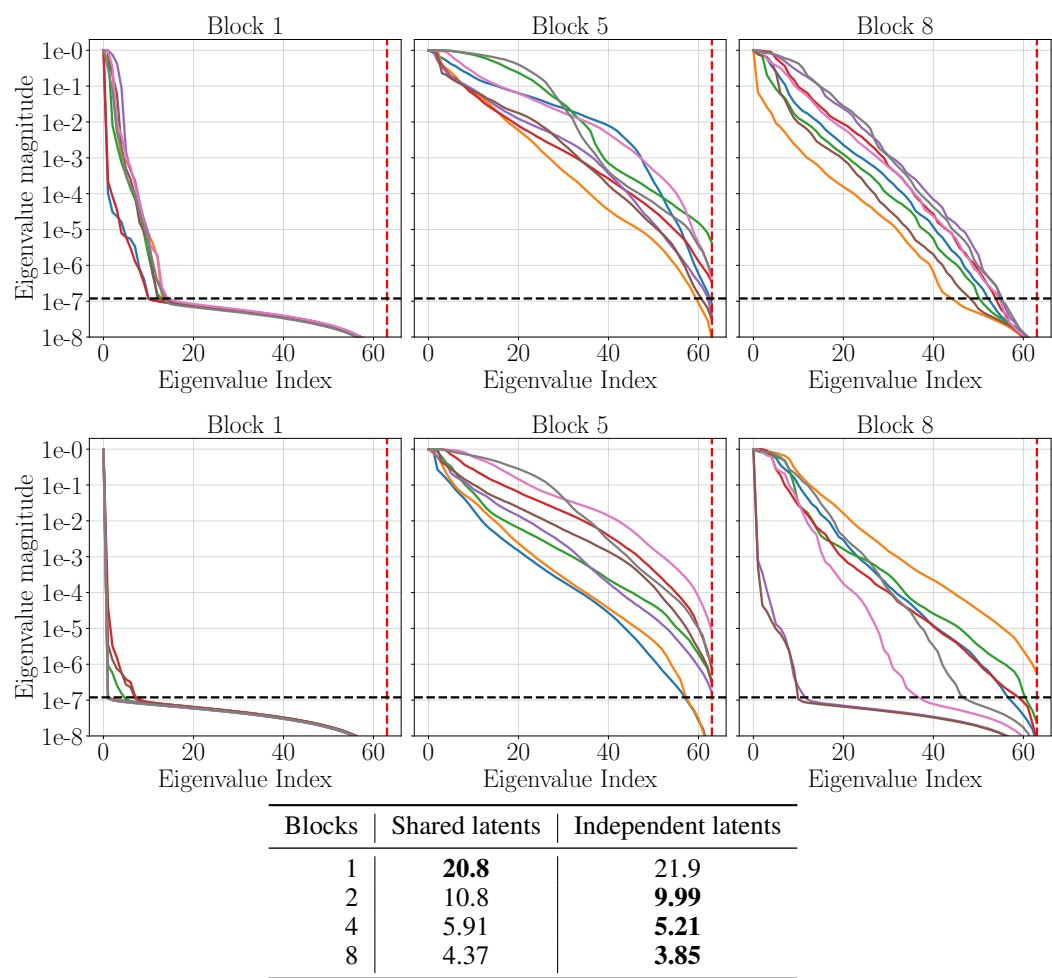

| Blocks | Shared latents | Independent latents |
|---|---|---|
| 1 | **20.8** | 21.9 |
| 2 | 10.8 | **9.99** |
| 4 | 5.91 | **5.21** |
| 8 | 4.37 | **3.85** |

Figure 13: **Ablation of shared vs. independent latent tokens across attention heads.** Each plot shows the $M=64$ nonzero eigenvalues of the head-specific communication matrices $W_h$ (Eq. 9) for FLARE with $B=8$ blocks and $C=64$ features trained on the elasticity dataset. When heads share a single latent sequence (top row), the eigenvalue spectra across heads are nearly identical, indicating similar learned low-rank subspaces. When heads use independent latent slices (bottom row), the eigenvalue decay varies noticeably across heads, reflecting more diverse low-rank structures. The accompanying table reports test relative errors for different depths $B$, showing that models with independent latents consistently achieve lower error.

## H BENCHMARK DATASET OF ADDITIVE MANUFACTURING SIMULATIONS

### H.1 INTRODUCTION & BACKGROUND

In metal additive manufacturing (AM), subtle variations in design geometry and process parameter selection may result in undesirable part artifacts or even costly build failures. Numerical simulations of laser powder bed fusion (LPBF), a popular additive manufacturing process, can be used to predict build failures, but this may take several minutes to hours depending on the part size.

Although AM enables the fabrication of a wide variety of new designs, the final product must nonetheless comply with the constraints of the underlying manufacturing process. Metal AM parts often have anisotropic and spatially varying material properties that depend on geometric features (such as overhang, or support structure) and fabrication process parameters (such as laser energy density, hatch spacing, layer thickness, and raster path). As-built parts with thin features often suffer distortions when manufactured with the LPBF process, which uses a high-power laser to selec-

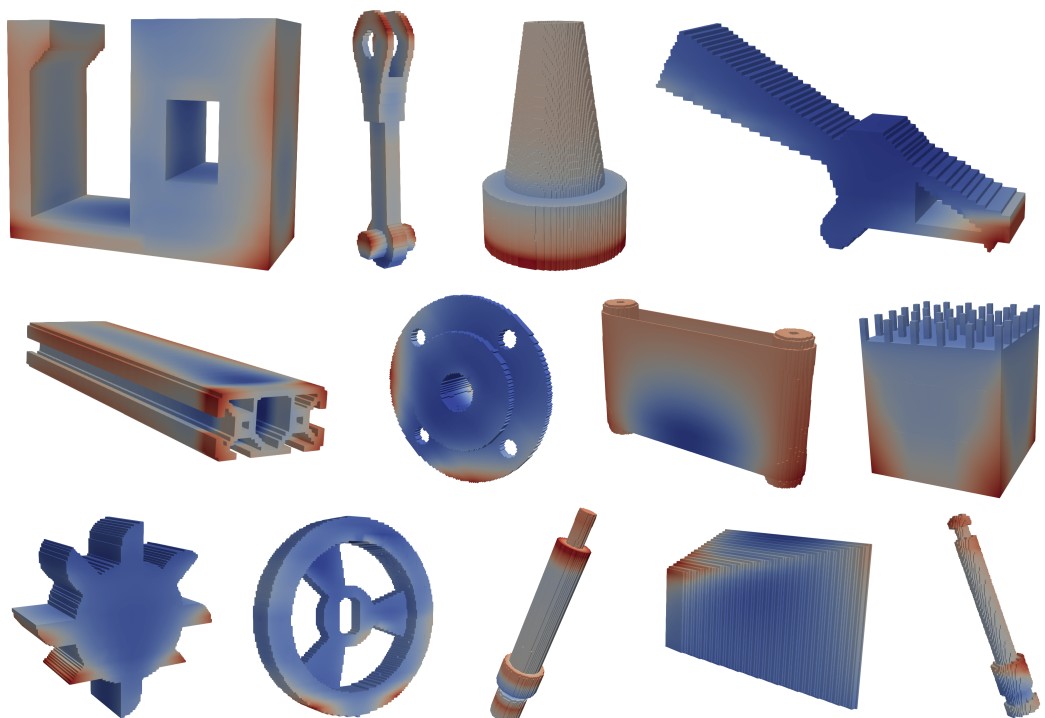

Figure 14: We simulate the LPBF process on selected geometries from the Autodesk segmentation dataset (Lambourne et al., 2021) to generate a benchmark dataset for AM calculations. Several geometries are presented in this gallery. The color indicates $Z$ (vertical) displacement field.

tively melt and fuse metal powder, layer-by-layer, to create complex, high-precision parts. Thermal stresses, termed *residual stresses* (RS), accumulate in LPBF-fabricated parts as a result of rapid thermal cycling due to laser exposure. These stresses can be severe to the point of inducing localized plastic deformations or delamination. As such, due to these *residual deformations*, the final shape of the part may deviate from the designed geometry.

We present a high-fidelity thermomechanical RS calculation data set on the Fusion 360 segmentation dataset, a publicly available dataset of complex 3D geometries (Lambourne et al., 2021). Numerical solvers for simulating the LPBF build process perform expensive quasi-static thermo-mechanical equations,

$$\underbrace{\rho C_p \frac{\mathrm{d}T}{\mathrm{d}t} = \boldsymbol{\nabla} \cdot k\Delta T(\boldsymbol{x}, t) + Q(\boldsymbol{x}, t),}_{\text{thermal transport}} \qquad \underbrace{\boldsymbol{\nabla} \cdot \boldsymbol{\sigma} = 0 \qquad \boldsymbol{\sigma} = \boldsymbol{C}\boldsymbol{\varepsilon}_e,}_{\text{stress equilibrium}} \tag{25}$$

layer-by-layer within a finite element framework (Denlinger et al., 2014; Liang et al., 2019; Autodesk, 2025), and are integrated in commercial software products (Autodesk, 2025). These calculations take several minutes to hours, making them prohibitively expensive for part design scenarios that can involve hundreds of evaluations.

Ferguson et al. (2025) introduced a dataset of LPBF simulations in which multiple finite element calculations were performed on a collection of 3D shapes. That dataset, however, was restricted to relatively coarse meshes with approximately 3,500 points per mesh. The present work extends this line of research by considering larger and more refined meshes, with up to 50,000 grid points.

### H.2 DATASET GENERATION

To generate a dataset of LPBF simulations, we employ Autodesk NetFabb (Autodesk, 2025), a commercially available software tool for numerically simulating RS and associated physics. We begin with the Fusion 360 segmentation dataset (Lambourne et al., 2021), and scale each shape to lie within

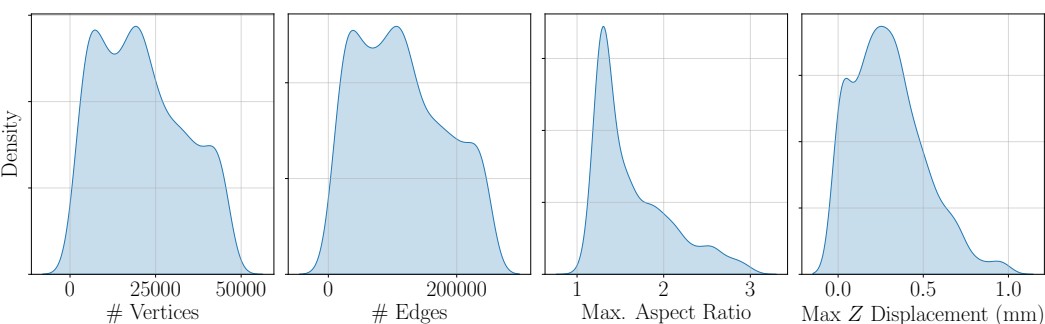

Figure 15: Summary of LPBF dataset statistics.

$[-30, 30] \times [-30, 30] \times [0, 60]$ mm such that it rests atop the build plate at $z \in [-25, 0]$ mm; the parts are not otherwise rotated or transformed. The simulation is then carried out for the Renishaw AM250 machine and *Ti-6Al-4V* material system deposited with $40$ $\mu$m thickness. Other parameters are left as their default nominal values and no support structures are added (Ferguson et al., 2025).

In AM, the material is deposited layer by layer, and ideally, a high-fidelity simulation would model each layer individually. However, this can be computationally expensive, especially for builds with hundreds or thousands of layers. Layer lumping simplifies this process by combining multiple physical layers into a single *lumped* computational layer. In our calculations, NetFabb applies layer-lumping with a lumped layer thickness of 2.5 mm.

NetFabb re-meshes the geometry to contain axis-aligned hexahedral elements before simulating the build process. Then, NetFabb generates a thermal and a mechanical history for each part corresponding to lumped layer deposition steps during the build. We obtain the displacement, elastic strain, von Mises stress, and temperature fields evaluated at all nodal locations throughout the build. NetFabb also provides field values after the part has cooled down and detached from the build plate.

### H.3 BENCHMARK TASK

To evaluate neural surrogate models, we target a field prediction task that is both central to our dataset and broadly relevant to the AM community: predicting residual vertical ($Z$) displacement. In LPBF, each layer is fused by a laser and followed by a recoater blade that spreads powder uniformly across the build area (Reijonen et al., 2024). Overhanging features may cause vertical displacements that interfere with the blade's path, potentially leading to collisions. Predicting the $Z$-displacement field can therefore help identify risk of blade collision failures. Rapid estimation of displacement prior to a build, thus, is highly desirable for design troubleshooting, as severe distortion can render a part unusable. Moreover, accurate prediction of nodal displacements can help anticipate build failures (Ferguson et al., 2025).

Since full-scale LPBF simulations are computationally expensive—taking minutes to hours—a fast surrogate model offers a valuable alternative for accelerating AM design. Accordingly, we train our models to predict the $Z$-displacement at every node at the final time step.

More formally, the input to a neural surrogate model is the volumetric axis-aligned hexahedral mesh describing the geometry. This includes the point-coordinates (array of size $N \times 3$ where $N$ is the number of points) and, optionally, mesh connectivity information. The corresponding label is the $Z$ displacement value at each point (array of size $N \times 1$).

While this dataset focuses on a steady-state prediction problem, future iterations of this benchmark could involve learning dynamic surrogate models that track the time-history of stress and deformation fields during the build process.

### H.4 DATA FILTERING

The dataset contains a wide-varying range of shapes, making the data set general enough to train a strong data-driven field prediction model. Out of $\sim$27,000 shapes, 19,732 were successfully simu-

Table 6: Summary statistics of our proposed LPBF dataset.

|      | #Points | #Edges | Avg./ max aspect ratio | Max height (mm) | Max Displacement |
|------|---------|--------|------------------------|-----------------|------------------|
| **Mean** | 20,972 | 114,140 | 1.6421 / 1.6421 | 29.429 | 0.29526 |
| **Std.** | 12,476 | 68,308 | 0.43794 / 0.43794 | 23.246 | 0.21064 |
| **Min** | 736 | 2,860 | 1.0667 / 1.0667 | 0.60000 | 0.00048500 |
| **25%** | 10,229 | 56,208 | 1.2800 / 1.2800 | 7.8000 | 0.13827 |
| **50%** | 19,743 | 107,680 | 1.4733 / 1.4733 | 21.600 | 0.27075 |
| **75%** | 30,503 | 166,250 | 1.9072 / 1.9072 | 60.000 | 0.41962 |
| **Max** | 47,542 | 249,930 | 2.9932 / 2.9933 | 60.000 | 0.99777 |

lated. We analyze the first 3,500 successful simulations and filter them according to several statistics to design a balanced training and test set. For example, we limit the learning problem to meshes with up to 50,000 points and up to 300,000 edges. This is done to reduce memory usage which becomes a bottleneck when training on small GPUs. We also filter meshes that have high aspect ratio elements as the FEM calculation could be unreliable on highly distorted geometries. The statistics for the filtered data set are presented in Table 6 along with histograms in Figure 15. A gallery of successful simulations is presented in Figure 14.

### H.5 QUALITATIVE RESULTS FOR LPBF $Z$-DISPLACEMENT PREDICTION

In Figure 16, we present visualizations of ground-truth $Z$-displacement, predictions by FLARE, and the corresponding error. Across a set of representative geometries from the LPBF test set, FLARE produces displacement fields that are visually indistinguishable from the ground truth and capture both the global deformation patterns and localized high-gradient regions. Importantly, the error fields remain low-magnitude and spatially diffuse, with no systematic accumulation near edges or corners—indicating that the model does not rely on positional shortcuts or overfit to particular geometric artifacts. This suggests that FLARE successfully learns the dominant thermo-mechanical modes of deformation while only missing small higher-order variations that are difficult to resolve without substantially larger latent bottlenecks. These qualitative observations are consistent with the low average relative $L^2$ error and illustrate that FLARE maintains stable accuracy across a diverse distribution of part geometries.

## I LLM USAGE

LLMs assisted with copyediting (grammar/clarity) and with scaffolding boilerplate in our experimental scripts (e.g., debugging, refactors) inside an IDE with LLM support. LLMs did *not* generate ideas, architectures, analyses, results, or datasets. All LLM outputs were reviewed, edited, and tested by the authors, who take full responsibility for the content.

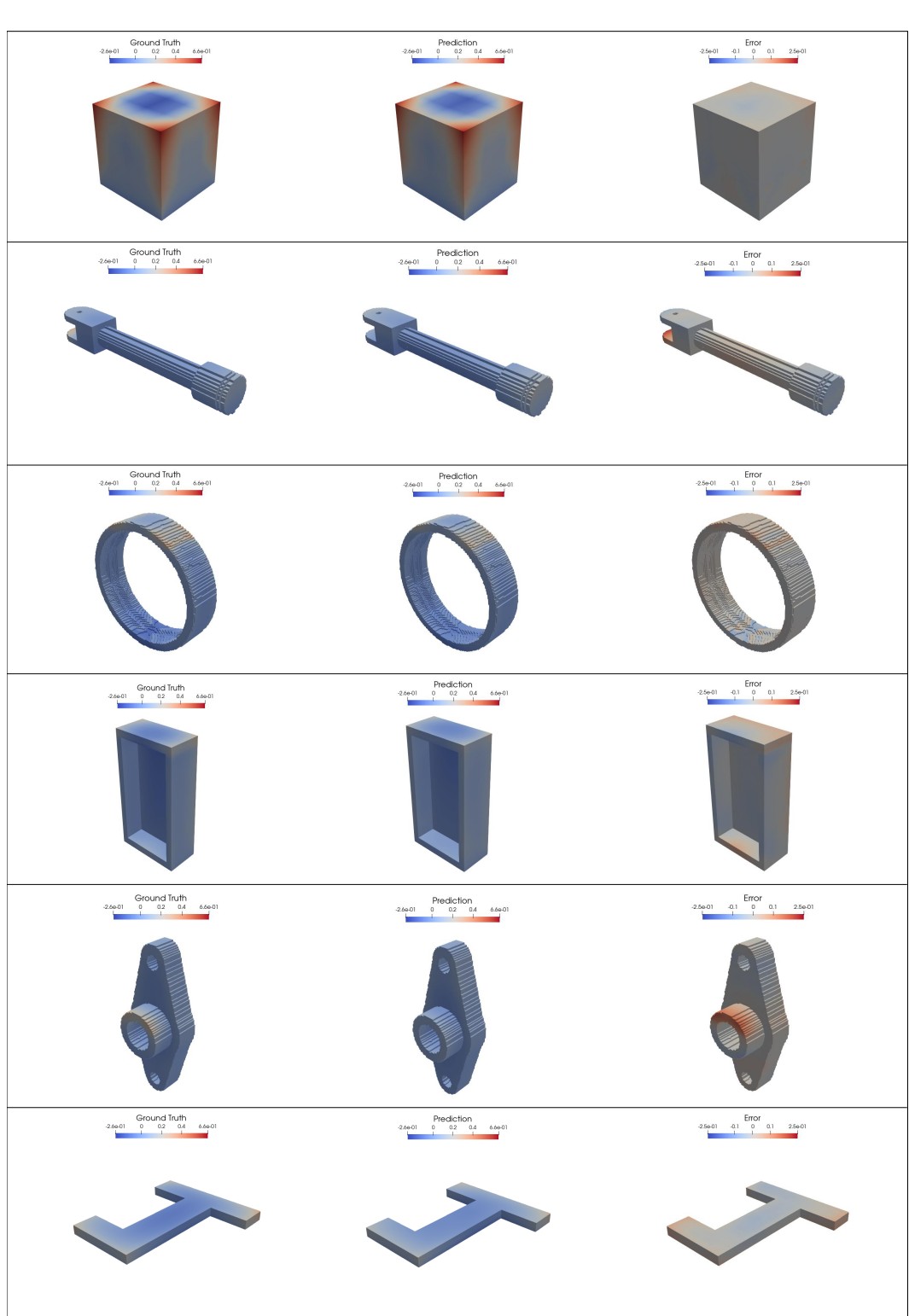

Figure 16: Qualitative results for FLARE on the LPBF $Z$-displacement prediction task. For each test geometry, we show the ground truth $Z$-displacement field, the model prediction, and the corresponding error (Ground Truth − Prediction).

