# OpenReview forum: "FLARE: Fast Low-rank Attention Routing Engine"
_ICLR.cc/2026/Conference — Submitted to ICLR 2026_

### Official Review · Reviewer_Be4h · 2025-10-19

**Soundness:** 2
**Presentation:** 3
**Contribution:** 2
**Rating:** 4
**Confidence:** 4

**Summary:**

This paper introduces FLARE, a linear-complexity attention mechanism designed to overcome the quadratic cost of traditional attention on large unstructured meshes. FLARE routes attention through learnable latent tokens, effectively forming a low-rank attention matrix that balances efficiency and accuracy. Experiments show that FLARE outperforms advanced PDE surrogate models across multiple benchmarks on accuracy and efficiency. The authors also release a manufacturing dataset to further research in scientific machine learning.

**Strengths:**

- This paper improves the accuracy and efficiency of Transformer-based PDE surrogates.

- The proposed FLARE framework is clearly described and easily implemented.

- This paper provides a new dataset that is meaningful for the community.

**Weaknesses:**

- The contribution of the tech is incremental and limited. This work neither tackles a new important scientific problem nor proposes a fundamentally new solution. Regarding to the tech proposed, the core idea of latent tokens has already been explored in prior works such as PerceiverIO, Transolver, LNO and etc. The main contribution is a slight modification of the attention formulation but it is hard to see the advantage of this modification comparing to other models like Transolver from the technical analysis. For example, the attention is operated on sequences with length $M$ in Transolver and on sequences with length $N$ and $M$ in Flare. Why is Flare more GPU efficient if $N \gg M$ ? Are the improvements mainly due to the use of the SDPA kernel? More technical analysis and comparison should be included. Otherwise, the paper reads more like an engineering optimization report rather than a scientific research paper.

- Following the above item, in Table 1 and Line 74, the authors claim that Transolver cannot utilize the SDPA kernel. This is inaccurate. From the official repo of Transolver++, the slice self-attention of Transolver can be naturally implemented by SDPA kernel. This operation is the same in Transolver++ and Transolver, not involving formulation optimization. So the comparison in Table 1 is misleading and not convincing.

- The dataset is also listed as a main contribution. But there is very limited content about the proposed dataset in the main context. The writing needs better arrangement to justify its value and meaning.

- Lack of visualizations. Given that this is a work focus on solving scientific problems efficiently, visualizations of predicted vs. ground-truth physical fields (e.g., stress or flow distributions) are essential. If the simulation states are not physical meaningful, the improvements in numerical accuracy or computational efficiency are of little significance.

**Questions:**

- In Figure 1, the peak memory usage of vanilla attention is lower than physical attention and flare. Is this reasonable? Did you use any optimization trick? If yes, all baselines should use same optimization tricks.

- In Table 2, baseline models have different parameter counts. Did you compare the efficiency of different models with comparable parameter counts? This is important for fair comparison, especially for efficiency.

- In Figure 4, with different $M$, the peak GPU memory usage is similar. Does this imply that the memory bottleneck is not the attention operation?

- There is only spectral analysis of Flare in Section 3.3 and Appendix B. Without the same analysis of other baselines like Transolver or LNO, it is hard to get the conclusion “FLARE benefits from head-wise diversity”.

---

> ### Author Response · Authors · 2025-11-18
>
> We thank the reviewer for their comprehensive feedback on our manuscript.
> We appreciate this opportunity to clarify our writing and strengthen the quality of our work.
> Based on your comments, we have
> 1. revised the Introduction and added a schematic to visually explain the difference between PercieverIO, LNO, Transolver and FLARE,
> 2. added two new ablations addressing your comments (Appendix G)
>
> Separately, but importantly, we have added a comparison against other efficient attention methods on the Long Range Arena Benchmark suite (Tay et al.) (Section 4.3, Tab. 2) where our method outperforms all other methods.
>
> ## Novelty of FLARE
>
> > The contribution of the tech is incremental and limited. This work neither tackles a new important scientific problem nor proposes a fundamentally new solution. Regarding to the tech proposed, the core idea of latent tokens has already been explored in prior works such as PerceiverIO, Transolver, LNO and etc. The main contribution is a slight modification of the attention formulation but it is hard to see the advantage of this modification comparing to other models like Transolver from the technical analysis.
>
> **On the contributions of our method:**
> We thank the reviewer for raising these important questions regarding the conceptual contribution of our method.
> The reviewer is correct that latent-token architectures have a long history, and we do not claim to introduce latency itself as a new idea.
> Our contribution lies instead in a specific set of architectural ideas, validated through targeted ablations, that collectively produce a qualitatively different attention design:
> 1. eliminating all latent-space self-attention,
> 2. allocating independent slices of latent tokens to each attention head, and
> 3. using many shallow encode-decode blocks rather than few deep latent transformers.
>
> Taken together, these yield a unified, low-rank self-attention operator that achieves superior accuracy to prior methods, while avoiding the limitations observed in prior low-rank attention models (e.g., dependence on sequence structure, poor scalability in Linformer).
> As demonstrated in our ablations (Appendix G) and empirical results (Fig. 3, Table 1, Table 2), this formulation consistently improves accuracy, and scalability across several challenging benchmarks, most notably **enabling end-to-end training on one-million-point unstructured meshes on a single GPU**.
> To the best of our knowledge, no prior work combines these ideas into a single latent-attention mechanism, nor reports comparable scaling behavior.
>
> > For example, the attention is operated on sequences with length $M$ in Transolver and on sequences with length $N$ and $M$ in Flare. Why is Flare more GPU efficient if $N \gg M$?
>
> Thank you for your question which gets to the heart of our approach.
> To process an input sequence of $N$ tokens,
> the computational complexity of
> Transolver is $\mathcal{O}(2MN + M^2)$
> and that of FLARE is  $\mathcal{O}(2MN)$ where both models employ $M$ latents.
> Our approach is more efficient for two reasons.
> First, we eliminate latent space self attention ($M^2$).
> Second, the encoding and decoding mechanism in FLARE can be written as cross-attention operations making them compatible with GPU efficient SDPA kernels.
> Therefore, even though the computational complexity is similar, FLARE is much more memory and compute efficient (Fig. 3).
>
> > Are the improvements mainly due to the use of the SDPA kernel? More technical analysis and comparison should be included. Otherwise, the paper reads more like an engineering optimization report rather than a scientific research paper.
>
> Thank you for your question.
> Our gains are not merely the result of SDPA kernel usage, but due to the three architectural ideas:
> (i) eliminating latent space attention which simplifies the architecture;
> (ii) employing independent parallel encoding-decoding pathways; and
> (iii) using shallow encode-decode blocks rather than deep latent transformers.
> Our approach reduces the bottleneck operations to two cross-attention operations per block, regardless of whether a fused kernel is used.
> The resulting architecture is simpler than prior latent-space models, which leads to superior accuracy despite allocating fewer model parameters (Table 1).
> Thus, SDPA is not the source of our accuracy gains.
>
> To ensure that each architectural component of FLARE is justified, we conducted five targeted ablations (three original, two new) examining runtime and memory scaling, the effect of depth versus latent dimensionality (Section 4.2, Fig 5, Fig 6), the role of head-wise parallel low-rank projections (Fig 10), the impact of latent-space self-attention (Fig. 11), and the necessity of independent per-head latent tokens (Fig. 12).
> Collectively, these studies isolate every design choice behind FLARE's formulation: its SDPA-compatible low-rank structure, repeated encode-decode blocks, elimination of latent self-attention, and head-wise latent diversity.

---

> ### Author Response · Authors · 2025-11-18
>
> ## Clarifications regarding scaling study
>
> > In Figure 1, the peak memory usage of vanilla attention is lower than physical attention and flare. Is this reasonable? Did you use any optimization trick? If yes, all baselines should use same optimization tricks.
>
> Thank you for the question. Yes, the behavior in Figure 1 is expected when using fused SDPA kernels.
> SDPA avoids materializing the full $N\times N$ attention matrix and instead computes attention in
> small tiles kept in registers, so the memory footprint of vanilla attention scales as
> $\mathcal{O}(N D)$ rather than $\mathcal{O}(N^2)$.This is why its peak memory is lower than both
> Transolver and FLARE.
>
> FLARE incurs slightly higher memory usage than vanilla attention because it must store the
> $M\times D$ latent sequence. However, this cost is small relative to the gains in compute efficiency
> from replacing quadratic $N^2$ attention with two $\mathcal{O}(NM)$ cross-attention operations.
>
> By contrast, Transolver's slicing/deslicing mechanism explicitly allocates the full
> $M\times N$ projection matrices, which dominates its memory usage and leads to out-of-memory failures
> at large $N$. Even though Transolver++ implements latent self-attention with SDPA, this optimization
> cannot be applied to the $\mathcal{O}(MN)$ slicing operations, which remain the primary bottleneck.
>
> > In Table 2, baseline models have different parameter counts. Did you compare the efficiency of different models with comparable parameter counts? This is important for fair comparison, especially for efficiency.
>
> Thank you for raising this point.
> For the time and memory measurements (Fig. 2 in the original manuscript; Fig. 3 in the revised version), **all models are matched in capacity**: we use the same feature dimension ($C=128$) and the same number of heads ($H=8$) for vanilla attention, Transolver, and FLARE.
> Transolver and FLARE are each evaluated at identical latent sizes $M\in\{128,512,2048\}$, ensuring that parameter counts and computational budgets are aligned.
> All methods use the same fused-attention backend wherever applicable.
> The corresponding implementation details can be found in the supplementary script `ablation/time_memory_bwd.py` (lines 244--251).
>
> > In Figure 4, with different $M$, the peak GPU memory usage is similar. Does this imply that the memory bottleneck is not the attention operation?
>
> Thank you for the question. Yes—this behavior is expected. In FLARE, the peak memory footprint is dominated by the $N\times D$ activations and their gradients, not by the latent dimension $M$. Because both cross-attention steps are executed with fused SDPA kernels, FLARE never materializes the $M\times N$ attention matrix; the only $M$-dependent allocation is the latent sequence $Z\in\mathbb{R}^{M\times D}$, which is negligible when $N=10^6$. As a result, varying $M$ changes runtime but has little effect on peak memory, which explains the nearly overlapping memory curves in Figure 4.
>
> ## **On spectral analysis and head-wise diversity**
>
> > There is only spectral analysis of Flare in Section 3.3 and Appendix B. Without the same analysis of other baselines like Transolver or LNO, it is hard to get the conclusion "FLARE benefits from head-wise diversity".
>
> We appreciate this question. We now include a **direct ablation** isolating head-wise latent diversity (Fig. 12). Models using a *shared* latent slice force all heads to learn nearly identical spectra and yield higher error. In contrast, *independent* latent slices produce diverse spectra and consistently lower error. This establishes the causal role of per-head latent independence.
>
> Regarding spectral comparisons with Transolver or LNO: these methods apply nonlinear latent-space transformations, so there is no single linear attention operator whose spectrum is meaningful. Thus, cross-model spectral comparisons are not well-defined; instead, controlled within-FLARE ablations provide the appropriate evidence.

---

> ### Author Response · Authors · 2025-11-18
>
> ## **Clarification regarding Transolver**
>
> > Following the above item, in Table 1 and Line 74, the authors claim that Transolver cannot utilize the SDPA kernel. This is inaccurate. From the official repo of Transolver++, the slice self-attention of Transolver can be naturally implemented by SDPA kernel. This operation is the same in Transolver++ and Transolver, not involving formulation optimization. So the comparison in Table 1 is misleading and not convincing.
>
> Thank you for your question.
> We would like to respectfully push back on this point.
> Please let us explain:
>
> The SDPA call used in Transolver++ corresponds to *latent-space self-attention*
> (see https://github.com/thuml/Transolver_plus/blob/main/models/Transolver_plus.py#L79),
> which operates on an $M$-token latent sequence and therefore has $\mathcal{O}(M^{2})$ cost.
> While this operation can indeed leverage SDPA, the resulting speedups are limited because
> the dominant cost in Transolver/Transolver++ arises from the **slicing/deslicing** steps
> that map between $N$ input tokens and $M$ latent tokens at $\mathcal{O}(MN)$ cost.
> These encode/decode operations  require explicitly materializing $M\times N$ projection
> matrices and thus cannot be implemented with fused SDPA kernels. In the following comment, we annotate the code for transolver++'s GitHub repo to illustrate this point.
>
> In contrast, FLARE expresses both the encoding and decoding operations *entirely*
> as SDPA-compatible cross-attention calls. This avoids materializing the $M\times N$
> projection matrices and enables all $\mathcal{O}(MN)$ bottleneck computations to benefit
> from fused-attention kernels. This distinction is what yields the strong scaling behavior
> observed in Fig. 3 and Fig. 5.
>
> ## **On the LPBF dataset and visualizations.**
> > The dataset is also listed as a main contribution. But there is very limited content about the proposed dataset in the main context. The writing needs better arrangement to justify its value and meaning.
>
> > Lack of visualizations. Given that this is a work focus on solving scientific problems efficiently, visualizations of predicted vs. ground-truth physical fields (e.g., stress or flow distributions) are essential. If the simulation states are not physical meaningful, the improvements in numerical accuracy or computational efficiency are of little significance.
>
> Thank you for this helpful suggestions.
>
> Appendix H documents the LPBF dataset (including visualizations, workflow, statistics, and splits), and Sec. 4.1 evaluates all models on it. The dataset is complete and will be released upon acceptance. We also added **new qualitative comparisons** of FLARE’s predicted displacement fields with ground truth, showing that FLARE faithfully captures both global deformation patterns and localized high-gradient regions, confirming that the predicted fields are physically meaningful.
>
> **On significance.**
> We emphasize that our core contribution is architectural: we achieve substantially higher accuracy than domain-specific surrogates (e.g., Transolver, LNO) with a simpler, and more scalable architecture.
> Most importantly, FLARE is the *first* attention-based surrogate capable of end-to-end training on **one million unstructured 3D points on a single GPU**, a scale of immediate relevance for industrial applications.
> This marks a meaningful practical milestone, not just an engineering refinement.
> Finally, our new Long Range Arena experiments, where FLARE achieves highest average accuracy across all models,
> (Section 4.3) show that FLARE's advantages extend well beyond PDE surrogates.
>
> We thank the reviewer again and are happy to revise further based on additional guidance.

---

> > ### Author Response · Authors · 2025-11-19
> >
> > ## Clarification regarding transolver (cont'd)
> >
> > > Following the above item, in Table 1 and Line 74, the authors claim that Transolver cannot utilize the SDPA kernel. This is inaccurate. From the official repo of Transolver++, the slice self-attention of Transolver can be naturally implemented by SDPA kernel. This operation is the same in Transolver++ and Transolver, not involving formulation optimization. So the comparison in Table 1 is misleading and not convincing.
> >
> > Below, we annotate the code from Transolver++'s github repo (https://github.com/thuml/Transolver_plus/blob/main/models/Transolver_plus.py#L31-L83) to indicate that the method uses SDPA only for latent-space self attention and not slicing/deslicing (encoding/decoding)
> >
> > ```python
> > class Physics_Attention_1D_Eidetic(nn.Module):
> >     def forward(self, x):
> >         # B N C
> >         B, N, C = x.shape
> >
> >         #---------------------------------------------------------------#
> >         # Step 1: Slicing (N --> M tokens)
> >         # Allocates of [B H N M] slice_weights (encoding/decoding projection matrices) to GPU (memory bottleneck)
> >         # Lines 64 -- 74
> >         # In this code `G' is the latent dimension whereas our document uses `M'
> >         #---------------------------------------------------------------#
> >         x_mid = self.in_project_x(x).reshape(B, N, self.heads, self.dim_head) \
> >             .permute(0, 2, 1, 3).contiguous()  # B H N C
> >
> >         temperature = self.proj_temperature(x_mid) + self.bias
> >         temperature = torch.clamp(temperature, min=0.01)
> >         slice_weights = gumbel_softmax(self.in_project_slice(x_mid), temperature)
> >         slice_norm = slice_weights.sum(2)  # B H G
> >         dist_nn.all_reduce(slice_norm, op=dist_nn.ReduceOp.SUM)
> >         slice_token = torch.einsum("bhnc,bhng->bhgc", x_mid, slice_weights).contiguous()
> >         dist_nn.all_reduce(slice_token, op=dist_nn.ReduceOp.SUM)
> >         slice_token = slice_token / ((slice_norm + 1e-5)[:, :, :, None].repeat(1, 1, 1, self.dim_head))
> >
> >         #---------------------------------------------------------------#
> >         # Step 2: Latent space self attention (M --> M tokens)
> >         # Implemented with SDPA
> >         # Lines 76 -- 79
> >         #---------------------------------------------------------------#
> >         q_slice_token = self.to_q(slice_token)
> >         k_slice_token = self.to_k(slice_token)
> >         v_slice_token = self.to_v(slice_token)
> >         out_slice_token = F.scaled_dot_product_attention(q_slice_token, k_slice_token, v_slice_token)
> >
> >         #---------------------------------------------------------------#
> >         # Step 3: Deslicing (M --> N tokens)
> >         # Lines 81 -- 83
> >         #---------------------------------------------------------------#
> >         out_x = torch.einsum("bhgc,bhng->bhnc", out_slice_token, slice_weights)
> >         out_x = rearrange(out_x, 'b h n d -> b n (h d)')
> >         return self.to_out(out_x)
> > ```
> >
> > In contrast, FLARE performs encoding and decoding by SDPA calls without materializing the $M \times N$ projection matrices.
> >
> > **Figure 2:** PyTorch code for multi-head token mixing operation in FLARE.
> >
> > ```python
> > from torch import nn
> > import torch.nn.functional as F
> >
> > class FLARE(nn.Module):
> >     def forward(self, x, return_scores: bool = False):
> >
> >         # x: [B N C]
> >
> >         q = self.latent_q.view(self.num_heads, self.num_latents, self.head_dim) # [H M D]
> >         q = q.unsqueeze(0).expand(x.size(0), -1, -1, -1) # [B, H, M, D]
> >
> >         k = rearrange(self.k_proj(x), 'b n (h d) -> b h n d', h=self.num_heads) # [B H N D]
> >         v = rearrange(self.v_proj(x), 'b n (h d) -> b h n d', h=self.num_heads)
> >
> >         #--------------------------------------------#
> >         # Encode
> >         #--------------------------------------------#
> >         z = F.scaled_dot_product_attention(q, k, v, scale=1.0)
> >
> >         #--------------------------------------------#
> >         # Decode
> >         #--------------------------------------------#
> >         y = F.scaled_dot_product_attention(k, q, z, scale=1.0)
> >         #--------------------------------------------#
> >
> >         y = rearrange(y, 'b h n d -> b n (h d)')
> >         y = self.out_proj(y)
> >
> >         return y
> > ```

---

> ### Comment · Reviewer_Be4h · 2025-11-27
>
> Thanks for your detailed response. However, I still have following concerns that are not well addressed:
>
> - I suggest that the authors use different colors to indicate the modified parts in the revised version. As reviewers, we are not as familiar with the manuscript as the authors are, and it is difficult for us to identify which parts have been added or removed without a detailed comparison.
> - I still cannot find a description of the generated benchmark dataset in the main text. Since you list “Benchmark dataset for additive manufacturing” as one of your contributions in the introduction, you should introduce this dataset in the main text, at least the scenarios, tasks, generation methodology, parameters, etc. The main text should be self-contained, and readers should be able to obtain all necessary information about your contributions without having to search through the appendix.
> - Visualizations of the physical tasks are also missing in the main text. As this work focuses on AI for PDEs (even though the main contribution is improving efficiency), readers should be able to understand what physical tasks you test and whether they are solved well. Numerical results alone are insufficient; sometimes numerical metrics look good, but the motion or evolution is not physically meaningful. Additionally, the visualizations should cover at least half of the tasks you solve, rather than only the LPBF dataset. (Refer to Transolver, which provides abundant visualization to helpfully justify its value on accuracy, physics and efficiency)
>
> Additionally,
>
> - I note that the attention weight $\text{softmax}(Q\cdot K^T)\in \mathbb{R}^{M\times N}$ and is similar to the slice weight in Transolver, are they equivalent? Additionally, Transolver claims that mesh points with close features will derive similar slice weights, which means they are more likely to be assigned to the same slice. Can this also be observed in the low-rank cross-attention step?

---

> ### Author Response · Authors · 2025-11-27
>
> Thank you very much for the continued discussion and for clearly outlining the remaining concerns.
> We address each point below.
>
> ### Marked‐up revision.
> Thank you for this helpful suggestion.
> We have included a marked-up version of the paper in the supplementary materials (see file `manuscript_markedup.pdf`).
> This annotated file highlights all additions, while the main updated manuscript remains unmarked to ensure compatibility with the automated diff that will be produced after the discussion period concludes.
> We hope this provides the clarity you were looking for, and we appreciate your careful reading of our work.
>
> ### Dataset description and visualization in the main text.
> We are currently working to reorganize the dataset section so that a concise, self-contained overview appears in the main text (the full placement in Appendix H was due solely to page-limit constraints).
> We also fully agree that visualizations are essential. We are actively preparing predicted vs.\ ground-truth visualizations for multiple PDE tasks.
> We will notify the reviewer as soon as these revisions are complete.
>
> ### Relationship between FLARE's attention weights and Transolver's slice weights.
> Thank you for raising this important conceptual question.
> These mechanisms are equivalent in the sense described below. In FLARE, the encode and decode weights are computed as follows:
> given
>
> $Q \in \mathbb{R}^{H \times M \times D} \qquad \text{(learned query)}$
>
> $K, V \in \mathbb{R}^{B \times H \times N \times D} \qquad \text{(from input)}$
>
> we compute encoding and decoding weights
>
> $S = Q \cdot K^T \in \mathbb{R}^{B \times H \times M \times N} \qquad \text{(attention scores)}$
>
> $
>     W_{\text{enc}} =\mathrm{softmax}(S)\in\mathbb{R}^{B\times H\times M\times N}\qquad\text{(encode weights)}\\
> $
>
> $
>     W_{\text{dec}} =\mathrm{softmax}(S^T)\in\mathbb{R}^{B\times H\times N\times M}\qquad\text{(decode weights)}.
> $
>
> With these weights, the FLARE module computes the output
>
> $
>     Z = W_\text{enc} \cdot V \in \mathbb{R}^{B \times H \times M \times D}\\
> $
>
> $
>     Y = W_\text{dec} \cdot Z \in \mathbb{R}^{B \times H \times N \times D}.\\
> $
>
> In Transolver, the decode weights are obtained via a learned projection as follows: given
>
> $
>     X, F \in \mathbb{R}^{B \times H \times N \times D} \qquad \text{(from input)},
> $
>
> transolver computes decoding weights with the projection matrix $W_\text{slice} \in \mathbb{R}^{D \times D}$ as
>
> $
> W_{\text{dec}}=\mathrm{softmax}(X \cdot W_{\text{proj}})\in\mathbb{R}^{B\times H\times N\times M}.
> $
>
> The equivalent encode weights are implicitly taken to be
>
> $
> W_{\text{enc}}=\frac{W_{\text{dec}}^\top}{\sum\nolimits_{n} W_{\text{dec}}[:, :, n, :] }\in\mathbb{R}^{B\times H\times M\times N},
> $
>
> so Transolver effectively performs
>
> $
>     Z =W_{\text{enc}} \cdot F \in \mathbb{R}^{B \times H \times M \times D} \\
> $
>
> $
>     Z =\mathrm{attention}(Z) \in \mathbb{R}^{B \times H \times M \times D} \\
> $
>
> $
>     Y=W_{\text{dec}} \cdot Z \in \mathbb{R}^{B \times H \times N \times D}
> $
>
> Thus the slice weights in Transolver correspond directly to FLARE's low‐rank cross‐attention weights, with the main difference that FLARE *removes* latent‐space self‐attention and keeps both steps SDPA‐compatible.
>
> ### Whether similar "grouping by features'' is observed in FLARE.
> Yes, this effect appears in FLARE as well.
> As discussed in Appendix F ("Latent tokens enable gather–scatter communication"), input tokens with similar key vectors produce similar dot‐product scores with a latent query and therefore accumulate in the same latent token during the encode step.
> The decode step then scatters those aggregated values back to the corresponding input regions.
> This is directly analogous to Transolver's observation but implemented via low‐rank cross‐attention rather than learned slice projections.
>
> Moreover, assigning each head its *own* latent token slice
> ($Q[h, :, :]$ for head $h$)
> allows different heads to learn distinct routing patterns.
> Our shared–vs–independent ablation (Appendix G) confirms this causally: shared latents collapse to nearly identical spectra and yield higher error, whereas independent latents show diverse spectral patterns and achieve consistently lower error.
>
> We thank the reviewer again for the careful reading and constructive guidance.
> We will implement all requested changes and are happy to revise further if there are additional suggestions.

---

> > ### Author Response · Authors · 2025-11-30
> > **Visualizations and dataset details in the main paper**
> >
> > Thank you very much for these constructive suggestions. We have revised the manuscript accordingly.
> >
> > # Dataset description in the main text.
> > We have reorganized the main paper to include a new Section 4 (“Benchmark Dataset for Additive Manufacturing”), which provides a self-contained summary of the LPBF benchmark. Due to the page limit, the detailed generation pipeline and full dataset documentation remain in Appendix H, but all essential information needed to understand the contribution is now included in the main text as requested.
> >
> > # Visualizations of physical tasks.
> > We have added a new figure (Figure 5) which provides qualitative visualizations (ground truth, FLARE predictions, and error fields) for several PDE tasks, not only LPBF. These examples illustrate the physical correctness and qualitative behavior of the model across multiple benchmark problems. We agree that such visual checks are important, and we have ensured that the visualizations cover more than half of the evaluated tasks, as suggested.
> >
> > We appreciate the reviewer’s guidance, and we believe the manuscript is now clearer, more self-contained, and more informative to readers.

---

### Official Review · Reviewer_nbxi · 2025-10-31

**Soundness:** 2
**Presentation:** 3
**Contribution:** 1
**Rating:** 4
**Confidence:** 4

**Summary:**

Self-attention is quadratic in sequence length, which makes it hard to apply to very large unstructured meshes. Authoers propose FLARE, a linear-complexity alternative. Each attention head routes global communication for N tokens through a learned latent bottleneck of M < N tokens. This induces a low-rank attention map (rank < M) and reduces cost to O(N·M). By choosing M, users can trade accuracy for efficiency.

**Strengths:**

1) Authors demonstrate a transformer based surrogate with global communication training directly on ~1M-point meshes on a single H100 80GB GPU using off-the-shelf fused attention kernels, and provide scaling curves. Scalability on experiments is something that has not gotten enough attention in the literature in this field, which makes these experiments the strong point.

2) They retrain all baselines (Perceiver IO, Transolver, LNO, etc.) under standardized splits, resolutions, hyperparameters, and model sizes. This makes the reported accuracy numbers more trustworthy.

**Weaknesses:**

1) FLARE’s low-rank attention via a latent bottleneck is directly linked to the attentions of Perceiver/Perceiver IO (iterative cross-attention into a fixed-size latent that scales linearly with input size) and Linformer (self-attention is low-rank; approximate it to get O(N) complexity). There are various methods along the same direction which makes the novelty of the concept unclear. If the novelty lies in O(N^2) -> O(N.M) then the concept is not new. If the mechanism by which this procedure happens (encode-decode) please see issues 2 and 3.

2) Authors do not run any meaningful ablations on the encode-decode mechanism which makes it unclear if this per-block encode decode is actually contributing to the results. The only ablations are minor (MLP depth, symmetry tie/untie), so the central architectural claim is not causally demonstrated. If the effect of encode-decode is not clear then do we fall back to the Preciever, or linformer style attention?

3) The following claim is vert strong " However, the latent bottleneck in PerceiverIO can limit accuracy as the model may discard fine-grained features if the number of latent tokens is too low. ". Preciever style model keep the original tokens [M x ...] and conduct cross attention to these at the beginning of each block. How does this discard information?

4) Authors claim giving each head its own latent slice is a key idea for expressivity, but never ablate a shared-latent variant. The evidence is descriptive spectra plots, not causal. Ablations are missing to show what would have been the effect of a shared latent token. This is especially important since it's not something that has been explored or adopted widely with attention mechanism in general.

5) The idea of weight sharing across cross attentions is also from Preciever paper so it's not fair to have the claim that PrevieverIO did not include the weight sharing (Table 1).

**Questions:**

Questions are formed as part of the weakness. Major concern is around the novelty of the work and the claims. I kindly request authors to focus on the novelty, and the claims (claims are specifically mentioned in the weakness) as they are clarifying the weaknesses.

---

> ### Author Response · Authors · 2025-11-18
>
> We thank the reviewer for their emphasis on FLARE's scalability and extensive comments on the core contribution of our method.
> We appreciate this opportunity to clarify our writing and strengthen the quality of our manuscript.
> Based on your comments, we have
> 1. revised the Introduction, and added a schematic of different efficient attention methods, and
> 2. added two new ablations addressing your comments (Appendix G).
>
> We also present a comparison of FLARE against other efficient attention methods (Linformer, Performer, Linear Attention) on the Long Range Arena benchmark suite (Tay et al.) (Section 4.3) where FLARE outperform all other models.
>
> ## Novelty and empirical validation of FLARE.
>
> > FLARE's low-rank attention via a latent bottleneck is directly linked to the attentions of Perceiver/Perceiver IO (iterative cross-attention into a fixed-size latent that scales linearly with input size) and Linformer (self-attention is low-rank; approximate it to get O(N) complexity).
> > There are various methods along the same direction which makes the novelty of the concept unclear.
> > If the novelty lies in O(N^2) $\to$ O(N.M) then the concept is not new.
>
> **On the novel of our method:**
> The reviewer is correct in pointing out that the concept of latent bottleneck attention is not new.
> We have as such revised the introduction to emphasize that
> (i) the core novelty of our approach is not the latent tokens themselves, but the **structural simplification** (no latent self-attention) that reduces the computation to a **single, low-rank formulation** self-attention operator,
> and (ii) these modifications have a marked impact on the accuracy and scalability of FLARE.
>
> Specifically, we propose three key architectural ideas validated through systematic ablations:
> 1. eliminating all latent-space self-attention,
> 2. allocating independent slices of latent tokens to each attention head, and
> 3. using many shallow encode-decode blocks rather than few deep latent transformers.
>
> Taken together, these yield a unified, low-rank self-attention operator that achieves superior accuracy to prior methods, while avoiding the limitations observed in prior low-rank attention models (e.g., dependence on sequence structure, poor scalability in Linformer).
> To the best of our knowledge, the combination of low-rank formulation, head-wise latent diversity, and the absence of latent-space transformers distinguishes our work from all prior latent-attention models, and enables our superior accuracy and scaling results.
>
> **On ablations suggested by the reviewer:**
> We concur with the reviewer's comment that our original manuscript did not provide sufficient ablations to causally validate our claims.
> To ensure that each of these claims is causally validated, we describe five ablations (three original and two new):
>
> 1. **Time & memory scaling.**
> We show that FLARE exhibits true linear $\mathcal{O}(NM)$ scaling in runtime and memory.
> FLARE's SDPA-compativle formulation enables long-sequence training without materializing $M\times N$ weights.
>
> 2. **Blocks vs. latent tokens.**
> Across all PDE benchmarks—including the 1M-point DrivAerML dataset—accuracy improves consistently with more FLARE blocks, whereas increasing $M$ helps only on inherently high-rank tasks.
> This validates that repeated low-rank mixing (depth) is the most effective driver of global communication.
>
> 3. **Headwise parallel encoding-decoding pathways.**
> Varying the number of attention heads while keeping total width fixed shows that FLARE performs best with many small heads, supporting our central claim that multiple independent low-rank projection pathways (one per head) approximate richer attention patterns than a few large heads.
>
> 4. **(NEW) Latent-space attention vs. encoding/decoding blocks.**
> In response to the reviewer's comment, we have added a new ablation (Fig. 11) varying the number of latent-space blocks ($L_B$) and FLARE blocks ($B$) shows a clear trend: *latent-space self-attention worsens accuracy and increases compute time*, while adding more encode-decode blocks consistently improves accuracy. The best results occur at $L_B = 0$, directly validating FLARE's choice to eliminate latent-space self-attention.
>
> 5. **(NEW) Shared vs. independent latent tokens.**
> In response to the reviewer's comment, we have added a new ablation (Fig 12) comparing *shared-latent* and *independent-latent* variants that shows that shared latents force all heads to learn nearly identical spectra *and* consistently yield higher test error. To the contrary, independent latents produce diverse eigenvalue decay patterns and achieve lower error.
> We believe, these results suggest that head-wise latent independence is beneficial, supporting FLARE's use of parallel, head-specific low-rank pathways.
>
> Together with our strong results, we believe our approach does not simply inherit the ideas of Perceiver or Transolver.
> Instead, we propose a new self-attention block that outperforms previous approaches.

---

> > ### Author Response · Authors · 2025-11-18
> >
> > ## Clarifications Perciever, PerceiverIO
> >
> > > The following claim is vert strong " However, the latent bottleneck in PerceiverIO can limit accuracy as the model may discard fine-grained features if the number of latent tokens is too low. ". Preciever style model keep the original tokens [M x ...] and conduct cross attention to these at the beginning of each block. How does this discard information?
> >
> > Thank you for bringing this to our attention.
> > Our reading of the PerceiverIO paper (Jaegle et al., 2021) did not indicate the use of $N \to M$ cross-attention after encoding.
> > This was based on Figure 1 of (Jaegle et al., 2021), where cross-attention is absent after encoding, and on the following lines in Appendix A.1 of their paper:
> > *"We omit the repeated encoder cross-attends used in Jaegle et al. (2021) as we found these to lead to
> > relatively small performance improvements but to significantly slow down training;"*
> > and the following lines in Appendix E:
> > *"Perceiver IO uses cross-attention for encoder and decoder attention modules and uses self-attention for the latent processing modules."*
> >
> > However, your comment made us wonder what the impact of cross-attention would be on PerceiverIO.
> > We ran the ablation and the results are as follows.
> >
> > **Table 1:** Relative $L_2$ error ($\times 10^{-3}$) and parameter count (in parentheses) for different models across PDE benchmark problems.
> >
> > | **Model** | **Elasticity** | **Darcy** | **Airfoil** | **Pipe** | **DrivAerML-40k** | **LPBF** |
> > |-----------|----------------|-----------|-------------|----------|-------------------|----------|
> > | PerceiverIO w/o cross-attn (original) | 60.6 (1.87m) | 26.9 (1.87m) | 25.6 (1.87m) | 8.81 (1.87m) | 760 (1.87m) | 560 (1.87m) |
> > | PerceiverIO with cross-attn (as suggested by reviewer) | 28.0 (1.87m) | 20.6 (1.87m) | 7.65 (1.87m) | 6.90 (1.87m) | 248 (1.87m) | 23.1 (1.87m) |
> > | **FLARE (ours)** | **3.38** (592k) | **5.10** (691k) | **4.28** (691k) | **2.85** (625k) | **60.8** (691k) | **18.5** (625k) |
> >
> >
> > As the reviewer astutely remarked, we see that PerceiverIO with cross-attention has significantly improved over PerceiverIO in its original form.
> > Nonetheless, we are happy to report that our method is able to outperform both variants on all test cases indicating that FLARE's encode-decode pathway captures richer global context than cross-attention in PerceiverIO.
> >
> > Based on the superior performance of PerceiverIO with cross-attention, we have updated its architecture everywhere in the paper.
> > We thus retract the claim regarding PerceiverIO losing information in the encoding step as that is clearly not true with cross-attention carrying information back to the latent space.
> >
> > > The idea of weight sharing across cross attentions is also from Preciever paper so it's not fair to have the claim that PrevieverIO did not include the weight sharing (Table 1).
> >
> > We thank the reviewer for bringing this to our attention.
> > Our reading of Perceiver (Jaegle et al., 2021) and PerceiverIO (Jaegle et al., 2021) indicated that those methods engage in weight sharing between sequential attention blocks in latent space.
> > From section 3.1 of (Jaegle et al., 2021):
> > *"Finally, in virtue of the iterative structure of the resulting architecture, we can increase the parameter efficiency of the model by sharing weights between the corresponding blocks
> > of each latent Transformer and/or between cross-attend modules. Latent self-attention blocks can still be shared if only a single cross-attend is used."*
> > Similarly, (Jaegle et al., 2021) makes several references to weight sharing in depth, but not to weight sharing between the encoder and decoder.
> >
> > To be clear, we do not claim that weight sharing is one of our contributions.
> > In fact, it has been used in previous works (Fig 1).
> >
> > In our original submission, our intention was to describe the coupling of encoding weights $(M \to N)$ and decoding weights $(N \to M)$.
> > Specifically in the context of an attention-based projector:
> > for $Q\in \mathbb{R}^{M \times D}$ (latent), and $K \in \mathbb{R}^{N \times D}$ (input),
> > the encoding and decoding weights are
> > $W_\text{encode} = \text{softmax}(Q \cdot K^T) \in \mathbb{R}^{M \times N}$,
> > $W_\text{decode} = \text{softmax}(K \cdot Q^T) \in \mathbb{R}^{N \times M}$
> > respectively.
> > We have as such reworded 'weight sharing' between encoder and decoder to 'encoder-decoder coupling' to avoid any misinterpretations.
> > Thank you for your helping in clarify this issue.

---

### Official Review · Reviewer_3k94 · 2025-11-01

**Soundness:** 3
**Presentation:** 3
**Contribution:** 3
**Rating:** 4
**Confidence:** 4

**Summary:**

In this paper, a linear self-attention mechanism is proposed for applying to large unstructured meshes. It is achieved by projecting the input sequence to a much lower-dimentional latent sequence. It enables training with 1 million points. The proposed model achieves superior accuracy compared to neural PDE models, including Vanilla Transformer, PerceiverIO, GNOT, LNO, and Transolver, on various datasets, including a newly constructed additive manufacturing simulation dataset.

**Strengths:**

The proposed method achieves sota PDE surrogate performance, while is also capable of handling geometries with a million points.

**Weaknesses:**

I think it is mandatory that in the experiments, this work also compares with other efficient attention methods proposed in general domains.

**Questions:**

Is there any design in the proposed method specifically for neural PDE scenarios? What if the proposed method applies to domains other than surrogate PDE solvers, e.g., LLM training?

---

> ### Author Response · Authors · 2025-11-18
>
> We thank the reviewer for their thoughtful feedback and their strong emphasis on the scalability of our model.
> Please see our response to your questions below.
>
> > I think it is mandatory that in the experiments, this work also compares with other efficient attention methods proposed in general domains.
>
> Thank you for raising these important points.
> To demonstrate the generality of FLARE, we added experiments on the Long Range Arena (LRA) benchmark suite (Section 4.3), which spans diverse long-context tasks in language, vision, and reasoning.
> We find that **FLARE achieves the highest average accuracy across all LRA tasks**, outperforming several efficient-attention baselines.
>
> **Table 1:** Accuracy (%) of different transformer models on Long Range Arena benchmark tasks (Tay et al., n.d.).
> The best result (highest accuracy) is **bold** and the second best is *italic*.
>
> | Model | **ListOps** | **Text** | **Retrieval** | **Image** | **Pathfinder-32** | **Avg** |
> |-------|-------------|----------|---------------|-----------|-------------------|---------|
> | Vanilla attention | **36.70** | *64.93* | *77.18* | 38.02 | 70.52 | *57.47* |
> | Linear attention | 17.15 | **66.00** | 71.84 | 09.86 | **75.00** | 47.97 |
> | Linformer | **36.70** | 53.00 | 64.72 | **41.88** | 70.09 | 53.28 |
> | Norm attention | 17.10 | 63.08 | 76.07 | 36.94 | 70.15 | 52.67 |
> | Performer | 35.90 | 64.21 | 68.42 | 35.36 | 53.83 | 51.54 |
> | **FLARE (ours)** | *36.15* | 64.00 | **77.30** | *40.96* | *71.91* | **58.06** |
>
> > Is there any design in the proposed method specifically for neural PDE scenarios? What if the proposed method applies to domains other than surrogate PDE solvers, e.g., LLM training?
>
> While our primary focus is large-scale PDE surrogate modeling, FLARE itself is not physics-specific, and consists only of repeated low-rank encode–decode attention blocks.
> Applying FLARE to next-token prediction and large-scale language modeling is a natural extension, and preliminary LLM-style modifications are currently under active development.
> Given FLARE's strong performance on both PDE tasks and LRA, we are optimistic about its applicability to other domains, and we consider such investigations an exciting direction for future work.
>
>
> ## References
>
> Tay, Yi, Mostafa Dehghani, Samira Abnar, Yikang Shen, Dara Bahri, Philip Pham, Jinfeng Rao, Liu Yang, Sebastian Ruder, and Donald Metzler (n.d.). "Long Range Arena: A Benchmark for Efficient Transformers". In: International Conference on Learning Representations.

---

### Official Review · Reviewer_wDJN · 2025-11-01

**Soundness:** 2
**Presentation:** 3
**Contribution:** 3
**Rating:** 6
**Confidence:** 4

**Summary:**

This paper introduces FLARE, a novel low-rank self-attention mechanism designed for efficient neural surrogate modeling of partial differential equations (PDEs) on point clouds or unstructured meshes. By leveraging known properties of rank-deficient matrices, the authors employs a linear complexity attention mechanism with a learnable latent bottleneck to further improve accuracy and efficiency on PDE surrogate modeling tasks. Furthermore, FLARE enables end-to-end training on unstructured meshes with one million points without distributed computing or memory offloading and proposes a new a 3D field-prediction benchmark dataset named LPBF.

**Strengths:**

1、The idea of low-rank self-attention is intersting. This wrok provides a good explanation on architectures design for  the neural operator domain, especially classical works and methods based on Transformer architectures.
2、A new dataset, LPBF, is proposed, making a valuable contribution to the advancement of research in this field.

**Weaknesses:**

1、The experimental datasets do not include one-dimensional or time-dependent PDE problems. It is recommended to add experiments demonstrating the model’s generality, for example by including shallow-water and reaction-diffusion equations from PDEBench as time-dependent cases.(Takamoto, Makoto, et al. "Pdebench: An extensive benchmark for scientific machine learning." Advances in Neural Information Processing Systems 35 (2022): 1596-1611.)
2、Since Mamba also achieves linear computational complexity, please compare and evaluate the proposed model against Mamba-based architectures, such as MambaNO and LaMO, highlighting their respective advantages and limitations.(Zheng, Jianwei, et al. "Alias-free mamba neural operator." Advances in Neural Information Processing Systems 37 (2024): 52962-52995.)(Tiwari, Karn, Niladri Dutta, and N. M. Krishnan. "Latent Mamba Operator for Partial Differential Equations." arXiv preprint arXiv:2505.19105 (2025).)

**Questions:**

1、The “Criteria” column in Table 1 lists several evaluation standards, but their specific meanings are not clearly explained. Please provide detailed descriptions for each criterion.

---

> ### Author Response · Authors · 2025-11-18
>
> We thank the reviewer for their thoughtful feedback.
> Based on your comments, we have
> 1. presented a comparison against LaMO on PDE benchmarks, and
> 2. revised the introduction, and added supplementary information (visual and prose) explaining the criteria in Table 1.
>
> We have additionally added a comparison against other efficient attention methods on the Long Range Arena Benchmark suite where FLARE outperforms all other methods.
>
> > The experimental datasets do not include one-dimensional or time-dependent PDE problems. It is recommended to add experiments demonstrating the model's generality, for example by including shallow-water and reaction-diffusion equations from PDEBench as time-dependent cases.(Takamoto, Makoto, et al. "Pdebench: An extensive benchmark for scientific machine learning." Advances in Neural Information Processing Systems 35 (2022): 1596-1611.) 2. Since Mamba also achieves linear computational complexity, please compare and evaluate the proposed model against Mamba-based architectures, such as MambaNO and LaMO, highlighting their respective advantages and limitations.(Zheng, Jianwei, et al. "Alias-free mamba neural operator." Advances in Neural Information Processing Systems 37 (2024): 52962-52995.)(Tiwari, Karn, Niladri Dutta, and N. M. Krishnan. "Latent Mamba Operator for Partial Differential Equations." arXiv preprint arXiv:2505.19105 (2025).)
>
> We thank the reviewer for raising this point.
> We would like to clarify that the domain of interest in this paper is PDE surrogate modeling on large 3D point clouds and meshes.
> Many PDEBench baselines, in comparison, (FNO, Transolver, LaMO, MambaNO, Mamba) rely fundamentally on **regular 1D/2D grids** through convolutions, Fourier modes, or causal scanning, which do not transfer to unstructured domains.
> To this end, in Section 4.1 and Table 1, we evaluate Transolver **with and without** its convolutional front-end on structured-grid test cases (Darcy, Airfoil, Pipe) and observe that performance drops sharply once convolutions are removed indicating that physics-attention alone is not an adequate global communication mechanism.
> As expected, convolution performs much of the heavy lifting on PDEBench-style problems.
>
> In contrast, FLARE assumes no grid structure, uses no convolutions, and remains fully permutation-equivariant, yet outperforms all models (with and without convolution) across all benchmarks.
> We also include comparisons with LaMO in this response, in both with and without convolution settings, and observe the same trend.
> While MambaNO and LaMO are valuable baselines, MambaNO's implementation and experiments are not for unstructured grids, and LaMO modifies only Transolver's latent update and therefore inherits its core scalability bottleneck—the $\mathcal{O}(MN)$ slicing/deslicing step, which prevents scaling to the million-point unstructured meshes considered here.
>
> **Table 1:** Relative $L_2$ error ($\times 10^{-3}$) and parameter count (in parentheses) for different models across PDE benchmark problems.
> A backslash ($\backslash$) indicates that the model cannot be applied to the benchmark.
>
> | **Model** | **Elasticity** | **Darcy** | **Airfoil** |
> |-----------|----------------|-----------|-------------|
> | Transolver w/o conv (Wu et al., 2024) | 6.40 (713k) | 18.6 (713k) | 8.24 (713k) |
> | Transolver with conv (Wu et al., 2024) | $\backslash$ | 5.94 (2.8m) | 5.50 (2.8m) |
> | LaMO w/o conv (Tiwari et al., n.d.) | 4.25 (1.72m) | 8.72 (1.72m) | 6.59 (1.72m) |
> | LaMO with conv (Tiwari et al., n.d.) | $\backslash$ | 5.44 (3.76m) | 4.65 (3.76m) |
> | **FLARE (ours)** | **3.38** (592k) | **5.10** (691k) | **4.28** (691k) |
>
> > The "Criteria" column in Table 1 lists several evaluation standards, but their specific meanings are not clearly explained. Please provide detailed descriptions for each criterion.
>
> Thank you for pointing this out.
> In response, we have revised the Introduction to provide a clearer exposition of the architectural differences among PerceiverIO, Transolver, LNO, and FLARE, and to introduce a schematic comparison figure that visually highlights these distinctions.
> For completeness, we also include detailed explanations of each criterion in Figure 1 in Appendix A.
>
> ## References
>
> Tay, Yi, Mostafa Dehghani, Samira Abnar, Yikang Shen, Dara Bahri, Philip Pham, Jinfeng Rao, Liu Yang, Sebastian Ruder, and Donald Metzler (n.d.). "Long Range Arena: A Benchmark for Efficient Transformers". In: International Conference on Learning Representations.
>
> Tiwari, Karn, Niladri Dutta, NM Anoop Krishnan, and Prathosh AP (n.d.). "Latent Mamba Operator for Partial Differential Equations". In: Forty-second International Conference on Machine Learning.
>
> Wu, Haixu, Huakun Luo, Haowen Wang, Jianmin Wang, and Mingsheng Long (2024). "Transolver: A fast transformer solver for pdes on general geometries". In: arXiv preprint arXiv:2402.02366.

---

### Meta-Review · Area_Chair_qLNZ · 2026-01-06

**Summary:**

This submission received three ratings of 4 and one rating of 6, placing it below the acceptance threshold. The primary concerns consistently raised by the reviewers were limited perceived novelty and insufficient experimental validation.

In particular, both nbxi and Be4h emphasized that the proposed low-rank attention routing mechanism closely resembles existing approaches such as Q-Former and PerceiverIO.  Additionally, all reviewers pointed out missing ablations or experiments. Commonly cited gaps included the lack of systematic comparisons against a broader set of linear and efficient attention baselines, as well as the absence of large-scale evaluations outside the targeted PDE domain. The AC also agrees that at least a lab-scale ImageNet classification/generation or an large-scale LLM benchmark should be added.

While the ACs acknowledge that the proposed method may be new or less explored within the specific application domain considered by the authors, they agree with the reviewers that, in its current form, **the paper does not meet the novelty and experimental completeness bar expected at ICLR**. The AC estimated all reviewer will maintain their original score, leading to a rejection decision. The authors are encouraged to further polish the submission in line with the rebuttal discussion, strengthen empirical validation, and consider submitting the revised work to a future venue.

**Reviewer Concerns:**

| Reviewer | Addressed Concerns               | Outstanding Concerns                                                          |
| -------- | -------------------------------- | ----------------------------------------------------------------------------- |
| wDJN     | Partial clarification of scope   | Missing comparisons and broader validation                                    |
| 3k94     | -   | Lower performance than LinFormer in some tasks; Lack of experiments beyond the target domain                                  |
| nbxi     | Clarifications on design choices | Limited novelty relative to Q-Former and PerceiverIO; missing broad ablations |
| Be4h     | Some efficiency clarifications   | Incremental contribution; lack of large-scale experiments                     |

**Reviewer Scores:**

| Reviewer | Initial Score | AC Estimated Score | AC Reason |
|---------|---------------|--------------------|-----------|
| wDJN    | 6             | 6                  | -
| 3k94    | 4             | 4                  | Baselines and beyond target domain concerns remain |
| nbxi    | 4             | 4                  | Novelty concerns persist |
| Be4h    | 4             | 4                  | Beyond target domain concerns remain |

---

### Decision · Program_Chairs · 2026-01-26

Reject